# Review on Nanomaterial-Based Melamine Detection

**Muthaiah Shellaiah** [1]  **and Kien Wen Sun** [1,2,*]

1    Department of Applied Chemistry, National Chiao Tung University, Hsinchu 30010, Taiwan;
     muthaiah1981@nctu.edu.tw
2    Department of Electronics Engineering, National Chiao Tung University, Hsinchu 30010, Taiwan
*    Correspondence: kwsun@mail.nctu.edu.tw

**Abstract:** Illegal adulteration of milk products by melamine and its analogs has become a threat to the world. In 2008, the misuse of melamine with infant formula caused serious effects on babies of China. Thereafter, the government of China and the US Food and Drug Administration (FDA) limited the use of melamine of 1 mg/kg for infant formula and 2.5 mg/kg for other dairy products. Similarly, the World Health Organization (WHO) has also limited the daily intake of melamine of 0.2 mg/kg body weight per day. Many sensory schemes have been proposed by the scientists for carrying out screening on melamine poisoning. Among them, nanomaterial-based sensing techniques are very promising in terms of real-time applicability. These materials uncover and quantify the melamine by means of diverse mechanisms, such as fluorescence resonance energy transfer (FRET), aggregation, inner filter effect, surface-enhanced Raman scattering (SERS), and self-assembly, etc. Nanomaterials used for the melamine determination include carbon dots, quantum dots, nanocomposites, nanocrystals, nanoclusters, nanoparticles, nanorods, nanowires, and nanotubes. In this review, we summarize and comment on the melamine sensing abilities of these nanomaterials for their suitability and future research directions.

**Keywords:** food poisoning; melamine; milk products; nanomaterials; aggregation; nanodots and nanoparticles; composite materials; electrochemical detection; surface-enhanced Raman scattering; colorimetric recognition; fluorescent sensors; inner filter effect; fluorescence resonance energy transfer

## 1. Introduction

Recently, food safety has become a major issue due to the increase of occurrences of food poisoning [1–3]. For example, illegal adulteration of toxic materials, such as clenbuterol in meat and melamine in dairy product, leads to food poisoning and severe health problems [4–10]. In this light, melamine has been a well-known additive—with 66% nitrogen content—and is still utilized in the production of many plastics, adhesives, glues, fertilizers, plywoods, cements, cleansers, and retardant paints [11–15]. It has been illegally consumed in many dairy products due to its low toxicity, and hence leads to many health problems. Even though melamine is a low-toxic material, with a high concentration it may cause renal pathology and death of infants. Moreover, melamine can be hydrolysed to cyanuric acid in vitro, which forms an insoluble melamine–cyanurate complex and causes the formation of kidney stones and renal failure by obstruction [16–18].

On this track, several food poisoning incidents have signalled the need for control over illegal use of melamine. For instance, incidence of illness and death of pet animals was reported in North America in 2007 due to the formation of melamine–cyanurate crystals in the kidneys [19]. Similarly, nearly 300,000 infants were affected by an infant formula—contaminated with melamine—resulting in six deaths in 2008 [20]. The government of China limited the use of melamine of 1 mg/kg for infant formula and 2.5 mg/kg for other dairy products since then [21]. Subsequently, the World Health

Organization (WHO) also fixed the daily intake limit of melamine to 0.2 mg/kg body weight per day [22], whereas the US Food and Drug Administration (FDA) promulgated the allowed limit of melamine of 1 mg/kg in infant formula and 2.5 mg/kg for other dairy products [23].

To rectify the harmful effects of melamine, diverse analytical tactics have been developed towards its detection in dairy and food items. Among them, the chromatographic techniques, such as high-performance liquid chromatography (HPLC), ultra-performance liquid chromatography/tandem mass spectrometry (UPLC/MS/MS), and gas chromatography/mass spectrometry (GC/MS) [24–33], produced convincing results and proved their suitability for melamine determination. However, apart from exceptional sensitivity and precision, these chromatographic and mass spectral methods are time consuming with high operating cost and requires skilled operators. Moreover, sample preparations for these chromatographic techniques are difficult, therefore they do not meet the requirement of on-site applicability. Hence, the development of profitable and easily synthesizable/fabricable melamine recognition probes with real-time applicability becomes essential. Such demand has been witnessed by the publications on the melamine detecting probes as presented in Figure 1. The rapid increase in publication numbers from 2009 to 2018 demonstrates the importance of melamine identification in dairy products worldwide. Among these probes, nanomaterial-based melamine sensors are highly regarded due to their potential in food and related products. Hence, an overall review on nanomaterial-based melamine sensors is required to direct upcoming novel designs towards a highly efficient determination of melamine.

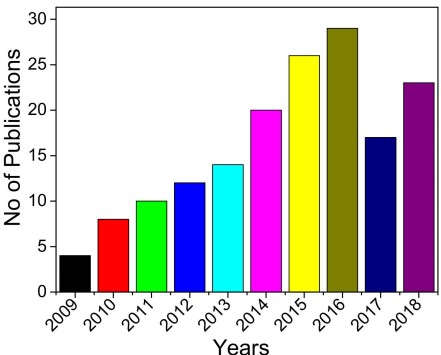

**Figure 1.** Numbers of publications on melamine sensors from 2009 to 2018 (adopted from Institute for Scientific Information (ISI) Web of Knowledge).

In this review, we survey and assess the nanomaterials-based melamine assays, especially covering reports on carbon dots, semiconductor quantum dots, nano-assemblies, nanoclusters, diverse nanocomposites, nanocrystals, nanoparticles, nanorods, and nanotubes, as well as other nanostructures, as illustrated in Figure 2.

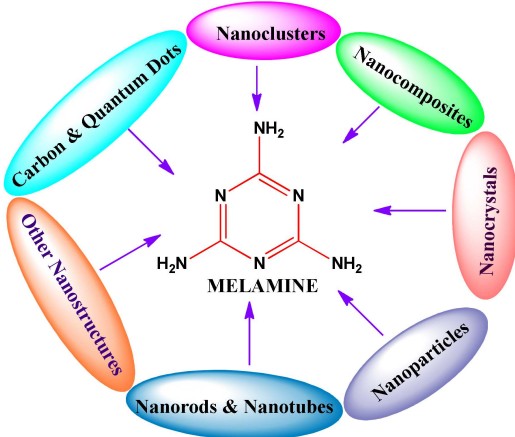

**Figure 2.** Schematic illustration of nanomaterial-based melamine determination.

## 2. Carbon and Quantum Dots for Melamine Detection

Due to their nano size (usually less than 10 nm), carbon dots (C-dots) have been utilized in numerous research fields. The unique properties in luminescence, biocompatibility, and low toxicity of C-dots allow them to be applied in a variety of analyte detections [34]. For example, Lei et al. utilized C-dots for the fluorescent "turn-on" determination of melamine via anti-quenching ability of $Hg^{2+}$ to C-dots [35]. Wherein, the melamine tends to coordinate with $Hg^{2+}$ through multi-nitrogen heterocyclic ring, which further leads to the anti-quenching and results to the fluorescence recovery of C-dots. Their work demonstrated a linear range from 1 to 20 μM (μM = micromole) and exhibited a detection limit (LOD) of 0.3 μM. Moreover, the obtained analytical recoveries in milk samples support reliability for real-time applications. Similar to the C-dot studies, Zhu and co-workers employed graphene quantum dots (GQDs) towards the selective assay of melamine in the presence of $Hg^{2+}$ ions [36]. In contrast to the C-dots, the melamine addition to GQDs in the presence of $Hg^{2+}$ ions showed emission quenching rather than enhancement via charge transfer mechanism as shown in Figure 3. The above work was authenticated by recoveries in milk samples with linear regression range between 0.15 to 20 μM and a LOD of 0.12 μM. Therefore, this method can be applied for melamine detection in real samples.

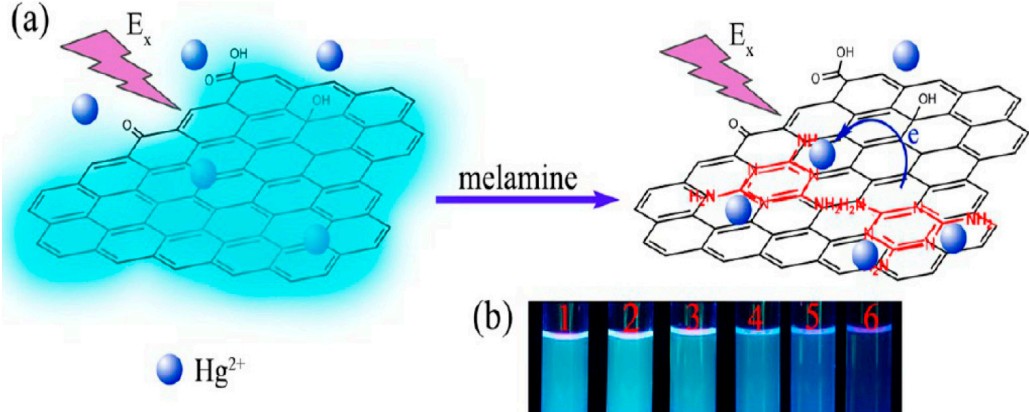

**Figure 3.** (**a**) Schematic illustration of melamine detection based on Fluorescence (FL) quenching of graphene quantum dots (GQDs) through charge transfer. (**b**) Optical photos of solutions of GQDs (Vial 1) and GQDs in the presence of melamine (Vial 2) or $Hg^{2+}$ (Vial 3), and of GQDs-$Hg^{2+}$ in the presence of 0.2, 0.8, 2.0 ppm melamine for Vials 4, 5, 6, respectively, taken under a 365 nm UV lamp (reproduced with permission from Reference [36]).

In this light, CdTe quantum dots were employed in the assay of melamine as noted next. In 2012, Zhang et al. publicized the use of thioglycolic acid-capped CdTe quantum dots (TGA-CdTe QDs) towards melamine detection [37]. The quantum dots were synthesized via microwave-assisted method, which discriminate the melamine via fluorescence quenching response. From the calibration plots, the linear regression of melamine detection was recognized as 79.2–793 μM with a LOD of 317 nM (nM = nanomole). This method shows high precision and accuracy during the melamine recognition in raw milk. Following the above work, Li and co-workers evaluated the TGA-CdTe QDs further towards the determination of melamine [38]. The estimated linearity of melamine ranged from 10 pM to 10 μM (pM = picomole) with a LOD of 5 pM. This method can be used to distinguish the melamine in alkaline aqueous solution as well.

Beside the aforementioned reports, molecularly-imprinted polymer (MIP) -capped CdTe quantum dots (MIP-CdTe QDs) were employed in the sensing of clenbuterol and melamine by Huy and co-workers [39]. The MIP-CdTe QDs were synthesized by radical polymerization process, which showed the luminescence quenching during the discovery of melamine. The linearity of melamine assay lies between 2.0–35 μM with a LOD of 0.6 μM. Notably, melamine in milk samples displayed more than 90% recovery. In a similar fashion, Xu and co-workers reported the molecularly-imprinted

CdTe QDs for ratiometric discrimination of melamine [40]. As shown in Figure 4, these MIP@CdTe QDs were synthesized in a single step and then utilized in ratiometric identification of melamine. The linear regression of melamine assay ranged from 100 to 800 nM with a LOD of 38 nM was reported. The luminescence recoveries in milk samples were established as 92~101%. This work can be used as a convenient, rapid, reliable and practical method for sensitive and selective fluorescence-based assay of melamine. However, the practicality of this study still needs to be further enhanced either by suitable modification or with combinations of other instruments

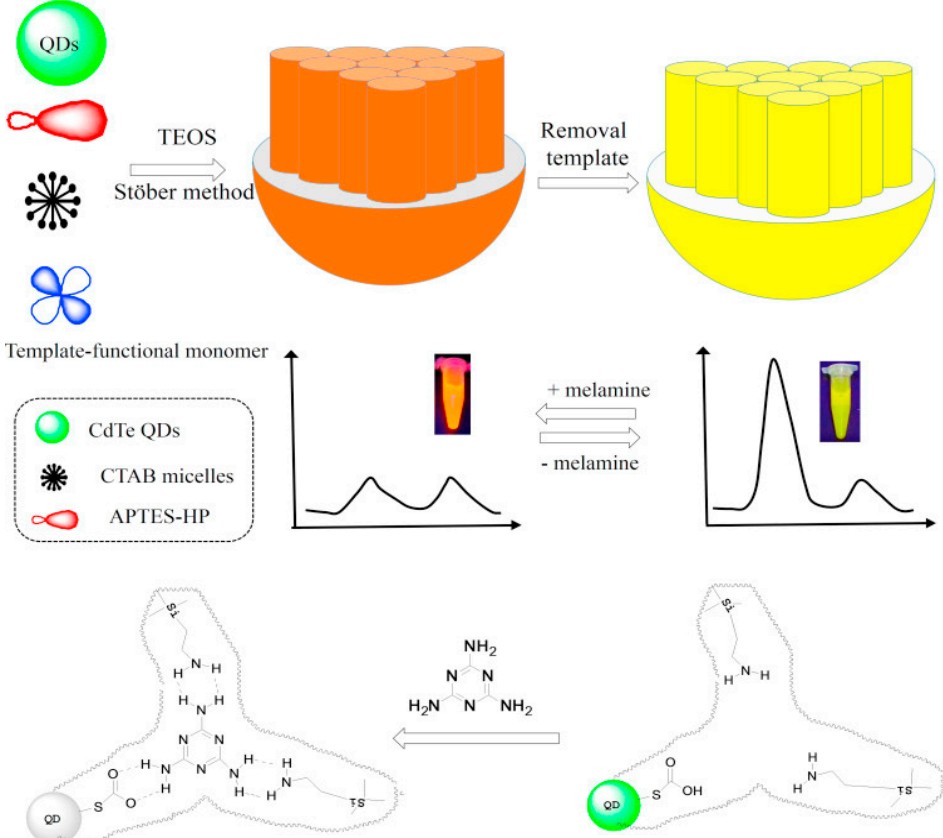

**Figure 4.** Schematic illustration for the one pot preparation of mesoporous structured ratiometric fluorescence of molecularly-imprinted polymers (MIPs) probe (reproduced with permission from Reference [40]).

At a later time, Zhang et al. presented a ratiometric fluorescent probe for visual detection of melamine (MEL) in milk samples [41]. The CdTe QDs with red emission were embedded within silica microspheres, and then the green emitting QDs were coated over silica microsphere surface as a shell. In addition, a molecularly-imprinted polymer (MIP) with binding site was placed on the shell for melamine recognition. During the melamine sensing the red-green fluorescence transformed into purely red via quenching of green emission due to the hydrogen bond interaction. The probe demonstrated a linearity between 396 nM–7.93 μM with a LOD of 103 nM. It also demonstrates the real-time applicability in milk samples and reveals the recoveries between 94.1~98.7% with 3.6–5.1% relative standard deviations (RSDs).

Cadmium and zinc sulfide quantum dots (CdS QDs and ZnS QDs) have also been used for melamine determination as described below. In 2012, Wang et al. established the fluorescent "turn-on" assay of melamine by thioglycolic acid-capped CdS quantum dots (TGA-CdS QDs) as illustrated in Figure 5 [42].

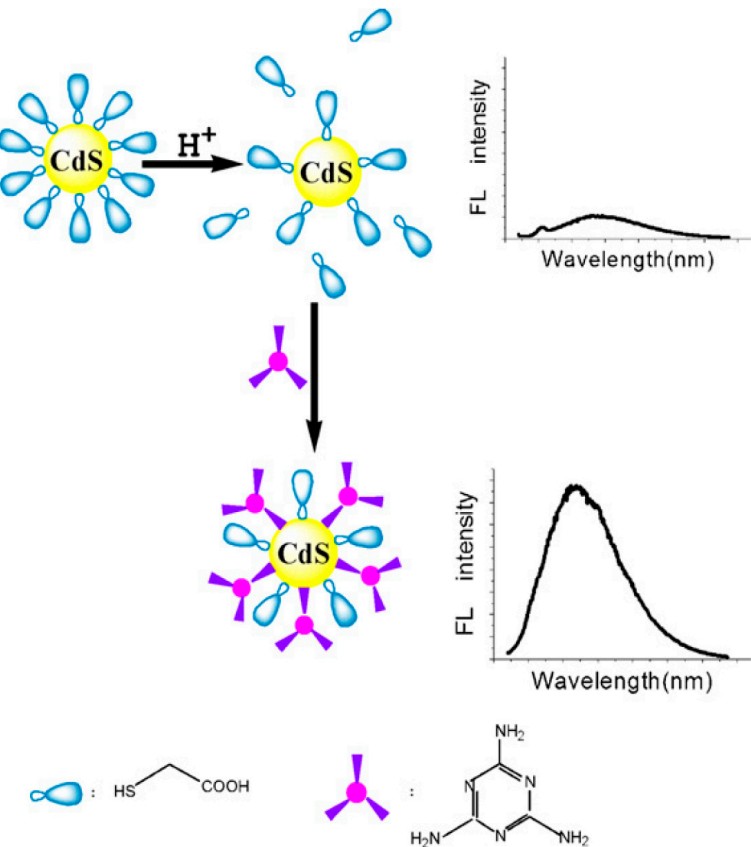

**Figure 5.** Scheme illustrating the interaction of melamine (MA) with thioglycolic acid (TGA) -capped cadmium sulfide quantum dots (CdS QDs) (reproduced with permission from Reference [42]).

Melamine was linearly identified between 2 nM–50 μM with an LOD of 1 nM under optimum conditions. This method is proposed as a highly sensitive and low-cost technique for melamine recognition in milk samples. Similar to the above proposal, the inner filter effect of gold nanoparticles on the fluorescence of CdS QDs was addressed in the detection of melamine in raw milk samples [43]. For this evaluation, the L-Cysteine-capped CdS QDs (L-Cys-CdS QDs) in the presence of citrate-stabilized Au NPs were utilized by Cao and co-workers to exhibit a linear regression between 396 nM–2.8 μM with a LOD of 135 nM. Apart from mild complications, this method can be employed towards rapid screening of melamine in milk products.

The water soluble europium ($Eu^{3+}$) doped ZnS quantum dots towards the detection of melamine in milk samples and infant formula at room temperature were exploited by Gong et al. [44]. The room temperature phosphorescence (RTP) of Eu(III)-ZnS QDs was quenched during the recognition process. The phosphorescence intensity at $\lambda_{ex}$ = 290 nm diminished linearly between 39.6 nM–3.96 μM with a LOD of 9.67 nM at pH 7.4. This technique was later applied in milk samples and displayed recoveries between 96~103% with a RSD of 1.2%. Demirhan and co-workers also proposed the RTP-based determination of melamine by means of L-cysteine-capped Mn-doped zinc sulfide (ZnS) quantum dots [45]. The RTP of this probe decreased linearly between 396 nM–3.96 μM of melamine with a LOD of 47 nM in 10 mM phosphate buffer (at pH 7.4). In dairy products, the recovery range of melamine was estimated as 96.3–104.7% with a RSD of 0.15%. These RTP-based melamine assay tactics can be effectively tuned to quantify melamine in infant formula, raw milk, cheese, yogurt and coffee creamer, etc.

Other quantum dots and its conjugates, such as $CuInS_2$ QDs, hapten-quantum dots bioconjugates (small molecules that stimulate an immune response while conjugated with a large carrier such as a protein are defined as haptens; in this case hapten = melamine) and secondary antibody

(Ab$_2$)-conjugated quantum dots (QDs) were also applied to melamine detection. A fluorescent "turn-on" based assay of melamine was demonstrated with CuInS$_2$ QDs [46]. Liu et al. described the recognition of melamine by CuInS$_2$ QDs, which had the initial fluorescence quenched by H$_2$O$_2$ oxidation. Only upon the addition of melamine, the fluorescence began to recover. This work displayed a linear concentration range of melamine from 10 nM to 10 μM with a LOD of 5 nM. This satisfactory recovery result on milk samples supports the real-time utility of this probe. However, the fluorescence quenching by oxidative step still makes the probe a complicated one.

Sanz-Medel's group developed the melamine–BSA-CdSe/ZnS QDs (BSA = Bovine Serum Albumin) conjugates and demonstrated their use in melamine detection via complementary optical spectroscopy and molecular mass spectrometry procedures [47]. In this work, a competitive immunoassay tactic was engaged for detection of melamine, where the melamine and the immunoprobe [QDs:Mel-BSA:EDC (conjugation buffer) at 2:1:1500 ratio] compete for the binding sites of the immobilized antibody. The Mel-BSA conjugates were developed by mixing Mel:BSA:EDC at 525:1:1700 ratio, which covered all the binding sites of BSA. Hence, melamine uses one of its NH$_2$ group and leaves other two NH$_2$ groups free to participate in the competitive immunoassay. This simple approach was also validated in melamine contaminated milk infant formula, which was in good agreement with other analytical methods. Without any sample pretreatment, a LOD 0.15 mg/kg was achieved, and hence can be applied for real-time monitoring of melamine contamination. An indirect fluorescence-linked immunosorbent assay (icFLISA) method based on secondary antibody (Ab$_2$)-conjugated quantum dots (QDs) was proposed for melamine detection by Wu and co-workers [48]. A LOD of melamine of 3.88 ng/mL was achieved, which was better than that of previous reports. Moreover, this method displayed recoveries between 80.85–110.54% with 2.82~8.82% RSDs in milk samples. The above QD-antibody-based immunoassay can be applied for rapid real-time screening of melamine in dairy products. However, these antibody-based QDs approaches still require sophisticated and costly biological instruments. The existence of other nitrogen enriched enzymes may considerably affect the selectivity of antibody-based QDs in melamine detection, which also extend the response time due to transport barrier.

## 3. Metal Nanoclusters Towards Melamine Determination

Similar to the carbon and quantum dots, luminescent metal nanoclusters were also applied in the screening of melamine contamination in milk products and infant formula. For instance, Dai et al. demonstrated BSA-stabilized gold nanoclusters (BSA-Au NCs) for the potential assay of melamine in raw milk and milk powder with good recoveries [49]. In this study, they proposed the fluorescent "turn-on" strategy for melamine recognition using anti-quenching capability of Hg$^{2+}$ to BSA-Au NCs. The emission from the Au NCs was quenched by Hg$^{2+}$ ions, then restored in the presence of melamine. Hence, this method can be applied for the identification of melamine adulteration in dairy products. This assay technique displayed linearity between 0.5–10 μM with a LOD of 0.15 μM. Moreover, its recovery range was between 93~102.5% with 2.69~4.52% RSDs in raw and powder milk samples. In this track, Yang and Liao's research groups developed the tiopronin-stabilized gold nanoclusters (TPN-Au NCs) for the discrimination of melamine by means of fluorescence quenching as shown in Figure 6 [50]. The probe displayed a linearity between 0.09–100 μM with a LOD of 32 nM. Potentiality in melamine detection of the probe was authenticated by its recoveries in spiked infant formulas, which was estimated as 92~102.2% and 1.14~2.80% RSDs.

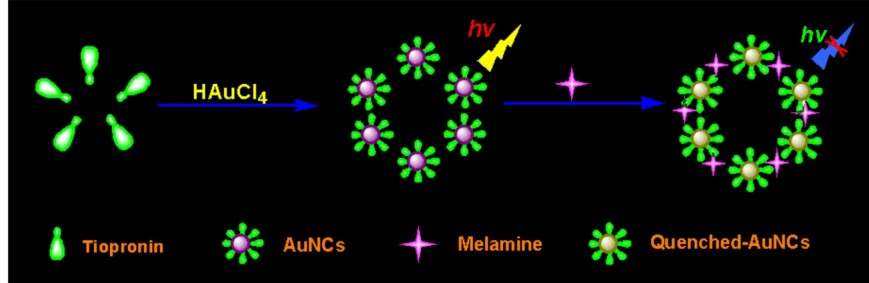

**Figure 6.** Scheme of the synthetic strategy for tiopronin-stabilized gold nanoclusters (TPN-AuNCs) and the principle of melamine sensing (reproduced with permission from Reference [50]).

On the other hand, Wang's research described the colorimetric recognition of melamine through reversing the inhibition of Hg(II) mediated light-triggered activity of horse-radish peroxidase (HRP) functionalized gold nanoclusters (HRP-Au NCs) [51]. In fact, the catalytic activity of HRP-Au NCs was restored by triazole ring of melamine via inhibition of the metallophilic interaction as mentioned in Reference [49]. It led to linear colorimetric "turn-on" assay of melamine between 0.2–15 μM with a LOD of 72 nM. Recoveries of this study in raw milk and milk powder were at the range from 98.5 to 101.5%. The above Hg(II) mediated sensing approach was further modified by Lin and co-workers [52]. They synthesized the BSA-Au NCs by microwave-assisted synthetic path and then applied in melamine detection as reported earlier. The melamine detection limit proposed in this work is 2.94 μM. However, apart from the synthetic route, this work describes the similar notion reported by Dai et al. Hence it cannot be attested as a suitable method for melamine recognition in dairy nutrients.

Melamine determination using glutathione-protected gold nanoclusters (GSH-Au NCs) was presented by Kalaiyarasan et al. through the fluorescence-based ratiometric assay [53]. Upon the addition of melamine, the photoluminescence (PL) intensity at 610 nm decreased along with enhanced ratiometric PL at 430 nm. The results were attributed to the hydrogen-bonding interaction between the melamine and AuNCs, which led to the aggregation of Au NCs. The proposed method applied in cow milk and infant formulas demonstrated recoveries between 94~97.1% with 1.24~3.95% RSDs. Furthermore, these GSH-Au NCs recognized the melamine with a LOD of 28.2 μM, hence can be applied in real-time screening of melamine. More recently, Lin and co-workers established the efficiency of egg-white protected gold nanoclusters (ew-Au NCs) through microwave technique [54]. However, the melamine detection by ew-Au NCs displayed the similar Hg(II) mediated approaches as described earlier [49,52]. The reported melamine detection has an LOD of 0.46 μM and recovery percentile range from 92.9 to 106% with RSD of 2.9%. Even though the author claimed the method innovative, however, their approach has already been established by earlier reports.

In 2012, Xu and co-workers reported the oligonucleotide-stabilized fluorescent silver nanoclusters (DNA-Ag NCs) towards "turn-on" recognition of melamine [55]. In that study, the linear melamine concentration fixed at 50 nM to 7 μM with a LOD of 10 nM was reported. The probe also identified the melamine in raw milk samples with recoveries from 100.4 to 107%. Similar to the functionalized Au NCs, hyper-branched polyethyleneimine-capped silver nanoclusters (PEI-Ag NCs) were employed in Hg(II) mediated melamine assay [56]. Qu and You's research groups developed the PEI-Ag NCs for Hg(II) induced melamine determination via "turn-on" fluorescent recovery. The linear detection range of melamine determination and LOD were estimated as 0.1 to 30 μM and 30 nM, respectively. Interestingly, recoveries from 96 to 101% were observed in food products. However, the idea and the approach of this work is the same as those of the Au NCs, except for the metallic cluster part. Hence its real-time applicability is questionable.

As an addition to the Ag NCs-based melamine sensors, Ren et al., discussed the utility of chromotropic acid and layered double hydroxides functionalized silver nanoclusters (CTA-Ag NCs/LDH) directed for fluorescence turn-on assay of melamine [57]. Ultrathin film consisting of CTA-Ag NCs and layered double hydroxide nanosheets (LDH) was fabricated via layer-by-layer (LBL)

assembly and was demonstrated successfully in the discrimination of melamine. In the presence of melamine, the fluorescence of ultrathin film displayed the "turn-on" response among other competitive species. Linearity of melamine quantification was observed from 0.03 to 0.1 μM with a LOD of 4 nM. This work uses the novel strategy of immobilization of metal nanoclusters into an inorganic matrix, which may enhance the upcoming research towards chemical and biosensing. As presented earlier [56], Hg(II) mediated melamine detection was conducted by Xie and co-workers [58]. In that study they also used the DNA-Ag NCs to expose the strong fluorescence recovery with melamine as shown in Figure 7. Linear melamine assay range was demonstrated between 0.2 to 4 μM with a LOD of 0.1 μM. This work is a duplication of earlier reports, hence cannot be considered as a suitable technique for real-time monitoring of melamine in dairy and food products.

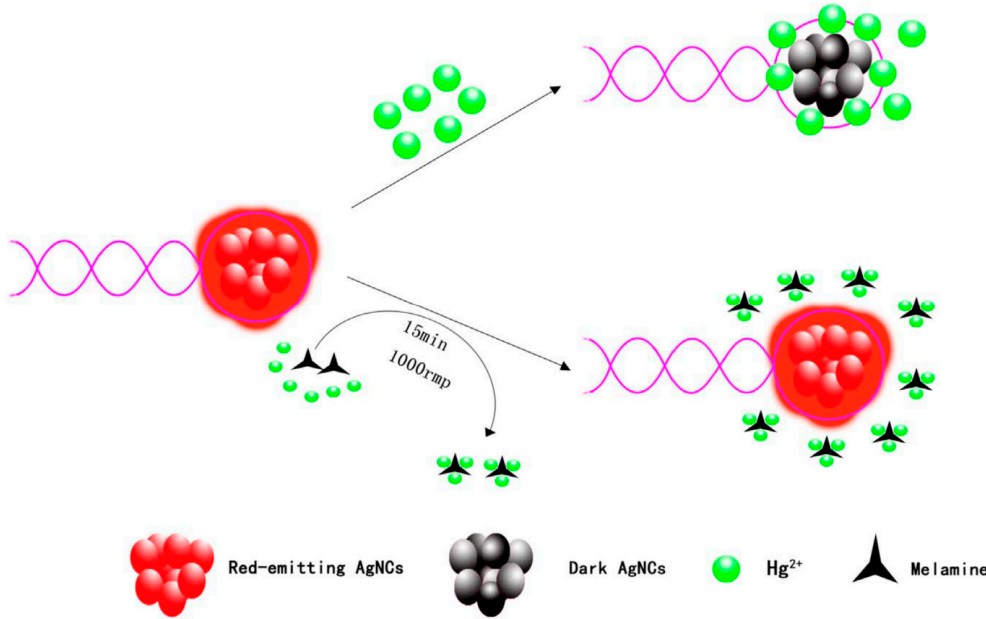

**Figure 7.** A fluorescence turn-on strategy for detecting melamine based on the coordination between melamine and $Hg^{2+}$ (reproduced with permission from Reference [58]).

Studies on the metallic-interaction-induced quantification of melamine were further stimulated by the recent reports from Li and Luo's research groups [59]. They employed the lipoic acid-stabilized silver nanoclusters (LA-Ag NCs), which emitted red fluorescence with Cu(II) and then facilitated melamine detection. The LA-Ag NCs displayed the aggregation-induced emission, which was then quenched with the addition of Cu(II) ions due to dispersion. Upon reaction with melamine, the emission restored to its original stage via formation of melamine-Cu(II) complex and then led to aggregation again. The LOD of melamine recognition was estimated as 174 nM; the recoveries and RSDs were oscillated from 98 to 113.3% and 3.12~5.30%, respectively. Although the work seems to be impressive, it can only be considered as a regular metallic-interaction-mediated melamine assay. On this path, Hou et al. presented silver nanocluster arrays over large-area silica nanosphere template for surface-enhanced Raman spectroscopy (SERS)-based assay of melamine with a lowered LOD of 100 nM [60]. Such a template-based assay approach can be directed toward real-time screening of melamine dairy and food products.

Luminescent copper nanoclusters were also engaged in melamine sensing as described next. Polythymine-stabilized copper nanoclusters were utilized for melamine recognition via enhanced fluorescence of Cu NCs [61] as seen in Figure 8. The linear melamine assay concentration was fixed between 0.1 and 6 μM with a LOD of 95 nM and the recoveries of spiked milk samples were established at 92 to 110% with 2~9.8% RSDs

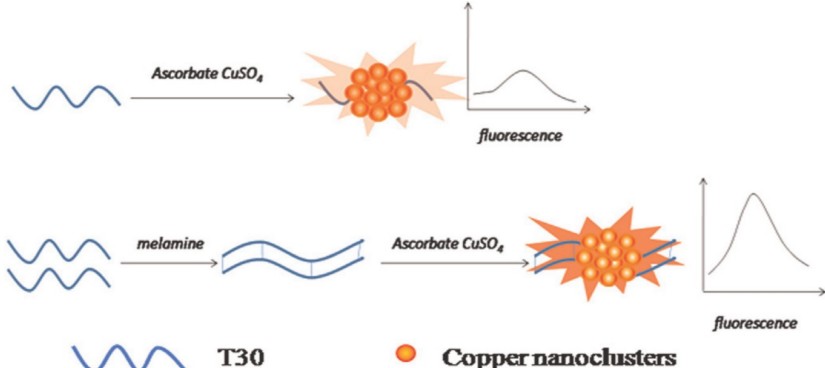

**Figure 8.** A scheme illustrating the melamine sensing based on the fluorescence enhancement of copper nanoclusters (reproduced with permission from Reference [61]).

## 4. Melamine Assay by Nanocomposites

Many nanocomposites have been reported as proficient candidates for melamine quantification, as described in this section. In 2011, Chen and co-workers presented the silver nanoparticle-decorated silver/carbon nanospheres (Ag/C/AgNps) as a composite candidate for SERS-based assay of melamine [62]. The effect of heavy metal ions on the determination of melamine was described in this report. The LOD of melamine was calculated to be as low as 50 nM, hence confirmed the good SERS activity of Ag/C/AgNps nanosphere composite. This is a unique technique and can be employed for real-time melamine screening in the presence of metal ions. Next, composites consisting of silver nanoparticles and glutathione capped zinc selenide quantum dots (GSH-ZnSe QDs) were exploited for melamine sensing in Cao's work [63]. As shown in Figure 9, the inner filter effect property (IFE) of Ag NPs is influenced by the fluorescence of GSH-ZnSe QDs and plays a vital role in the sensing process, wherein the initial non-aggregated Ag NPs are transformed to aggregated state in the presence of melamine, which provide the strong emission with GSH-ZnSe QDs. On the other hand, the mixture of Ag NPs and GSH-ZnSe QDs shows weak emission. The linear regression for melamine is observed between 7.93 to 285 nM with a LOD of 872 pM. This IFE-based sensing strategy is satisfactorily applied for melamine assay in raw milk and egg samples and has recoveries between 97.2 to 102.1% with corresponding RSDs among 0.7 to 3.7%. The above IFE-based tactic can be a suitable method for real-time scrutinizing of melamine in milk and food.

Regarding the composite nanomaterial-based analytical approach, a sandwich-type composite assembly is described as follows. Sarkar et al. presented the para-phenylenediamine (PDA) sandwiched between a nanostructured silver nanoparticle film and gold core−silver shell nanoparticles, which was further applied in SERS-based detection of melamine at femtomolar ($10^{-15}$ M) level [64]. These tailored sandwich-based approaches provide the innovative ways towards the development of SERS substrates which can be utilized in bio- and chemical sensors. However, the complications and the requirement of costly instruments to fabricate such substrates need to be overcome.

Towards the development of cost-effective SERS composite substrates for melamine recognition, Wu and Roy's research group developed the screen printed silver nanoparticles (Ag NPs) over a polyethylene terephthalate (PET) substrate [65]. Here the SERS-based melamine assay was established in liquid milk, but the clear linear ranges and limits were absent. Moreover, these substrates were also engaged to detect Rhodamine 6G (R6G) and Malachite green (MG) etc. Hence, more efforts are required in order to use these substrates towards melamine detection. A microfluidic chip consisting of an indium tin oxide (ITO) support modified with silver-gold nanocomposite (Ag–Au NCs) was demonstrated in the SERS-based determination of melamine by Wang and co-workers [66]. The results predicted a linear detection range of melamine from 10 nM to 0.1 mM with a LOD of 10 nM. On the other hand, this chip also shows sensitivity to 4-mercaptobenzoic acid (4-MBA) with a LOD of 0.1 nM. Therefore, it can act as an effective tool for melamine and biochemical analysis.

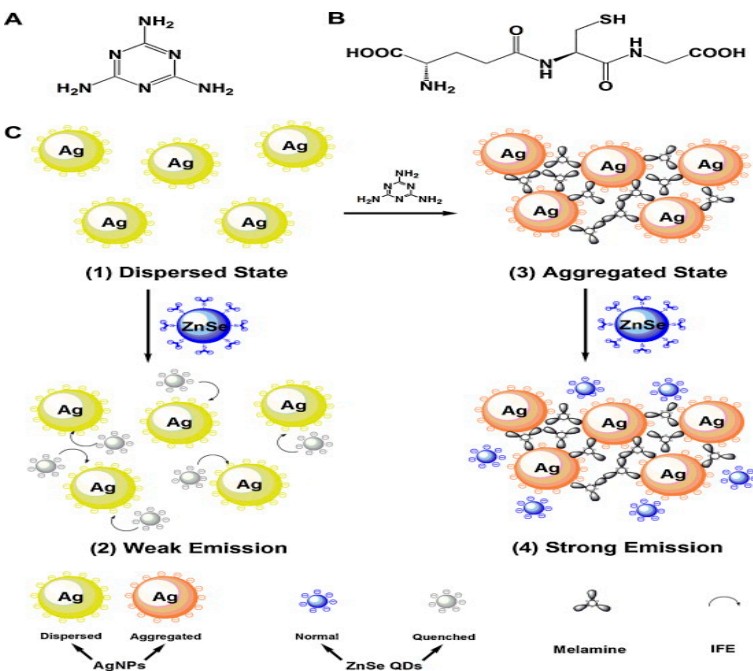

**Figure 9.** The molecular structures of melamine (**A**) and l-GSH (**B**); Schematic illustration of the "turn-on" fluorescent detection of melamine based on the inner filter effect (IFE) of AgNPs on ZnSe QDs (**C**) (reproduced with permission from Reference [63]).

Subsequently, Han et al. applied silver nanocontacts onto silica nanospheres (SiO$_2$@Ag) for SERS-based sensory studies [67]. The above composite substrate was employed in the SERS-based assays of various environment pollutants, such as thiram, melamine, and ethyl-parathion. Moreover, the estimated LOD for melamine assay was 1 nM. The SiO$_2$@Ag can be considered as one of the best SERS probes for multiple analyte detection. In 2011, Wang's research group demonstrated the SERS-based melamine assay using silver nanoparticle coated poly(styrene-co-acrylic acid) nanospheres (PSA/Ag NPs) [68]. Wherein, the composite [PSA/Ag-NPs/polyvinylpyrrolidone (PVP)] displayed a lower LOD on melamine detection near 1 mM in the presence of PVP. However, upon the removal of PVP, the LOD was enhanced to 100 nM. This work is also included the SERS-based sensing studies and can be further directed towards in vivo diagnostics and multimodal imaging [69].

An ultrasensitive colorimetric assay of melamine was delivered by Wang and co-workers through in situ reduction to form the carbon quantum dots (CQDs)-silver nanocomposite [70]. As shown in Figure 10, the solution in the presence of melamine changes its color. Under optimum conditions, the linearity of melamine assays was from 793 pM to 79.3 nM and 198 nM to 3.96 μM with a LOD of 62.6 pM. Furthermore, the work described the melamine recoveries in cow milk and milk powder, which were between 87.6~116% with 1.6~3.9% RSDs. This colorimetric detection approach can be used for real-time screening of melamine in food stuffs. In addition, polydopamine-glutathione nanoparticles and silver nanoparticles were employed to melamine discrimination based on the fluorescence resonance energy transfer (FRET) effect as demonstrated in Figure 11 [71]. In the presence of melamine, the Ag(I)−melamine complex was formed, which prevented the generation of Ag NPs to show the fluorescence "turn-on" response. In contrast, in the absence of melamine, Ag NPs were formed and hence no fluorescence was observed. The linear response to the concentration of melamine was from 0.1 to 40 μM with a LOD of 23 nM. The recoveries of melamine in milk, yogurt and infant formula stretched from 99.4 to 104.2% with 2.46~4.83% RSDs. Due to its simplicity in operation, this method can be directed towards real-time screening of melamine.

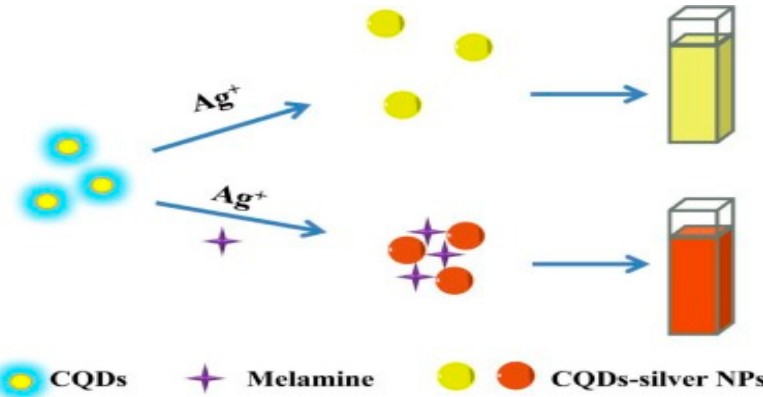

**Figure 10.** Schematic illustration of the colorimetric sensing of melamine based on in situ formation of CQDs-silver NPs (reproduced with permission from Reference [70]).

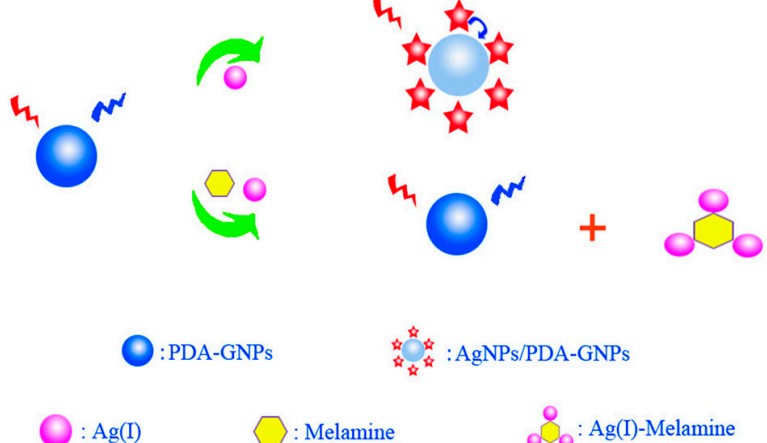

**Figure 11.** Schematic illustration of the sensing mechanism of melamine in the absence and presence of melamine (reproduced with permission from Reference [71]).

Manzoori et al. presented a beneficial effect of gold/silver alloy nanoparticles (Au/Ag NPs) on the chemiluminescence of permanganate-formaldehyde system in the presence of sodium dodecyl sulfate micelles [72]. Wherein, the trace of melamine inhibited the reaction of nanoparticles, which promoted the generation of Mn(II)* and enhanced the chemiluminescence intensity. Therefore, chemiluminescence of the above composite system was quenched considerably with melamine. The linear determination range for melamine was projected from 79.3 pM to 277.5 nM with a LOD of 63.4 pM. In powdered milk samples, the melamine recoveries were between 93.7~104.5% with 0.3~3.5% RSDs. Even though this work is considered as a good analytical approach, additional authentication is still required to enhance its real-time reliability. In this light, Sun and Zhang's report described the IFE of gold nanoparticles on the fluorescence of CdTe quantum dots, which led to fluorescence "turn-on" response [73]. Under optimized condition the detection limit of melamine was calculated to be 158.6 nM.

Yue and co-workers established the ability of gold nanoparticles deposited β-FeOOH nanorods (Au/β-FeOOH nanocomposites) in composite-based melamine recognition due to the promotion of symmetry-forbidden bands ($n - \mu^*$) of melamine [74]. Apart from detection mechanism, the assay of melamine in this work is insufficient, hence the reliability of this work is still questionable. Next, an electrode modified with gold nanoparticles and reduced graphene oxide (AuNPs/rGO) was demonstrated by Chen et al. towards the discrimination of melamine in food contact materials [75]. They used the hexacyanoferrate as a reporter in this electrochemical sensing. Its response was affected by the increase in the concentration of melamine. The linear regression of melamine assay was from

5 to 50 nM with a LOD of 1 nM and the sensing was also validated in food articles. The above electrochemical assay method can be effectively directed towards melamine detection in food and dairy objects.

Yang's research group engaged the studies on the functionalized Au-Fe$_3$O$_4$ nanocomposites for magnetic and colorimetric bimodal recognition of melamine [76]. They modified the Au-Fe$_3$O$_4$ nanocomposites with l-(2-mercaptoethyl)-1,2,3,4,5,6-hexanhydro-s-triazine-2,4,6-trione (MTT) and used them in bimodal sensing of melamine. In the presence of melamine, the color of the Au-Fe3O4@MTT NPs changes from red to completely colorless, hence can be used for naked-eye and on-site qualitative determination of melamine. Besides, the composite shows a linear response between 6 to 22 µM of melamine concentrations. However, this result still needs more work towards electrochemical sensors. Following this path, gold nanoparticles deposited on a graphene doped carbon paste was employed as an electrode for electrochemical sensing of melamine [77]. Melamine interacts with gold nanoparticles to supress the peak current. From 0.2 to 800 nM and 800 nM to 8 mM concentrations of melamine, the peak current displayed a good linear relationship with a LOD of 18 pM. The above quantitative melamine determination was validated by its recoveries in spiked milk samples, which were between 95.00~101.75% with 0.09~3.16% RSDs. Although this work is impressive, it requires a complicated fabrication processes.

Other than the earlier extensive works on the SERS-based discrimination [65–69], Neng's report also detailed the SERS-based discrimination of melamine by using Fe$_3$O$_4$/Au magnetic nanoparticles coated with 5-aminoorotic acid (AOA) as a substrate [78]. In their studies, the SERS substrate (Fe$_3$O$_4$/Au–AOA) and Rhodamine B (RhB) binded AOA (AOA–RhB) were engaged as a Raman reporter, which forms the supramolecular complex with melamine [Fe$_3$O$_4$/Au–AOA•••melamine•••AOA–RhB] via H-bonding as illustrated in Figure 12. From 19.8 to 119 µM, the linear regression of melamine was observed with a LOD of 19.8 µM.

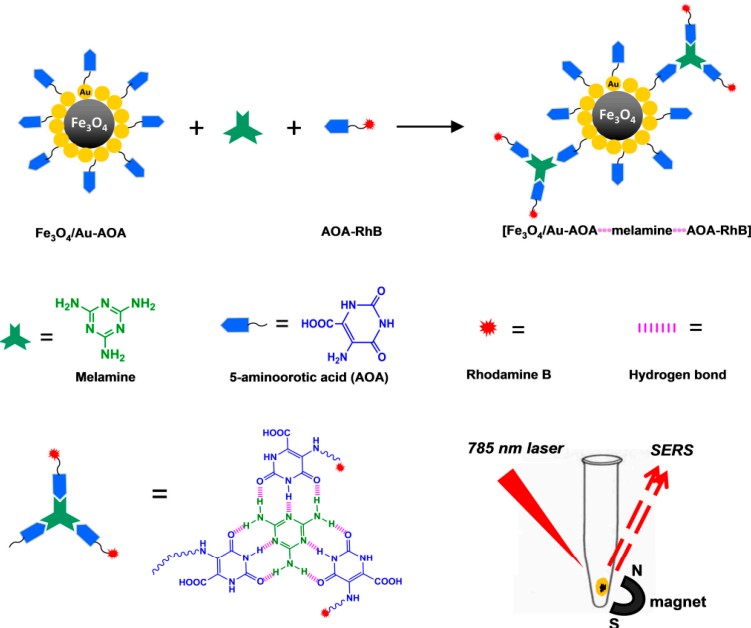

**Figure 12.** The H-bonding assisted [Fe$_3$O$_4$/Au–AOA•••melamine•••AOA–RhB] surface-enhanced Raman spectroscopy (SERS) matrix (reproduced with permission from Reference [78]).

Although this method possesses the complicated fabrication processes for Raman substrates, it avoids the sample pre-treatment steps, and hence is important in real-time monitoring of melamine. Rao et al. presented a novel electrochemical sensors by means of surface modified Glassy carbon electrode (GCE) with Au and polyaniline composites in the melamine screening, which enhanced the electrode sensitivity and sensor signal amplification [79]. During sensing processes, melamine

assembled on the Au@PANI via hydrogen bonds, which led to diverse electrochemical signal. LOD of melamine assay was estimated to be 1.39 µM. In spiked milk samples the recoveries were between 90.8~98.3% and 89.5~91.3%. This electrochemical method can be employed for on-site inspection of melamine in dairy and food products.

Niu and co-workers invented a novel dual-emission ratiometric fluorescence probe to determine melamine by combining the organic nanoparticles with aggregation-induced emission (AIE) characteristics and gold nanoclusters (Au NCs) through electrostatic interaction [80]. The above composite system (AIE-OFNs/Au NCs) with quenched emission at 625 nm detects the melamine through anti-quenching ability of Au NCs-Hg(II). Upon the addition of melamine, the fluorescence recovered back to its original state via Hg(II)-melamine link. The melamine recoveries in spiked powder milk samples were from 90 to 96.4% with 1.84~4.40% RSDs. The above record is an inclusion to the metal mediated melamine recognition. In the electrochemical-based determination of melamine, Dey et al. used a reduced graphene oxide–gold nanoparticle composite carbon paste as an electrode to recognize melamine by spotting the decrease in the oxidation of ferrocyanide [81]. This work also demonstrated good recoveries for melamine detection in milk powder and tap water. This work displayed a linear dynamic range from 5 to 160 µM with a LOD of 2.23 µM.

As an inclusion to SERS-based melamine discovery, a substrate containing curved-edge gold nanocubes (CENCs) and Au nanospheres (Au NSs) were demonstrated by Lv and co-workers [82]. The SERS substrate showed sensitivity to melamine with a LOD of 1 nM due to a tip–gap mesh structure. Mixture of carbon dots (C-dots) and gold nanoparticles (Au NPs) was reported towards fluorescence resonance energy transfer (FRET) based assay of melamine by Li's group [83]. The fluorescence of C-dots, which was quenched by Au NPs via FRET, recovered during the sensing process. The melamine recognition expressed the linearity from 50 nM to 500 nM with a LOD of 36 nM. These FRET-based melamine detection was also demonstrated through good recoveries in raw milk and milk powders, which were between 90.45~111.35% with 0.72~2.05% RSDs, and hence can be proficient in real-time monitoring of melamine.

On this track, Qu and co-workers presented a new ratiometric fluorescence probe via self-assembly of carbon nanodots and glutathione-stabilized gold nanoclusters (CNDs/GSH@Au NCs) for the selective discrimination of melamine [84]. Similar to their previous report [56], this study also demonstrated the anti-quenching ability of melamine to CNDs/GSH@Au NCs–$Hg^{2+}$, and hence can be considered as an addition to the metal mediated melamine assay. The linear melamine detection concentration was from 0.1 to 30 µM with a LOD of 29.3 nM. Rovina et al. described an electrochemical sensor composed of ionic liquid/zinc oxide nanoparticles/chitosan/gold electrode for melamine identification [85]. The authors used the (1-ethyl-3-methylimidazolium tri-fluoromethanesulfonate ([EMIM][Otf])) as ionic liquid, ZnO nanoparticles, chitosan nanocomposite membranes and gold electrode for melamine assay. The dynamic melamine detection concentration was from 0.96 pM to 0.96 µM with a LOD of 96 fM. The proposed method showed the melamine recoveries between 95.4~97.5% with 0.41~0.81% RSDs. Although this work shows the lowest LOD on melamine sensing, steps for fabrication of electrodes involve many complications, and hence much effort is still required.

Fluorescence energy transfer among CdTe-doped silica nanoparticles (CdTe@$SiO_2$) and gold nanoparticles (Au NPs) towards melamine determination was proposed by Gao and co-workers [86]. Wherein, upon the addition of Au NPs to CdTe@$SiO_2$, the fluorescence was quenched and then recovered in the presence of melamine. The melamine assay showed linearity between 7.5 to 350 nM with a LOD of 0.89 nM. This method also applied in raw milk and milk powder samples with exceptional recoveries between 97.4~104.1%, and hence can be utilized towards the screening of melamine in dairy products. On this path, a FRET-based melamine detection was discussed in Su's research report [87] which proposed the FRET between 3-Mercaptopropionic acid-capped CdTe QDs and Rhodamine B for the quantification of melamine. A linear quenching by melamine was achieved between 0.05~4.0 µM with an LOD of 0.01 µM. The above method was validated with melamine

recoveries in spiked milk samples, which were from 99.2 to 104% with 2.9~3.5% RSDs. However, continuing efforts are still required to improve the potential of this work.

A composite comprised of CdTe/CdS QDs and Au NPs was engaged in the assay of melamine by Zhao and collaborators as described in Reference [88], in which FRET between CdTe/CdS QDs and Au NPs led to the "turn-on" detection of melamine. Upon addition of melamine to the above composite (CdTe/CdS QDs and Au NPs), the fluorescence was enhanced linearly between 50 nM to 1 μM with a LOD of 30 nM. This method displays good recoveries for the melamine recognition in raw milk and milk products, which are from 90 to 101%. However, this tactic is only an inclusion to the FRET-based composite sensors for melamine. Next, an IFE-based melamine determination was demonstrated by Zhu et al. through a CdTe QDs and Au NPs composite system [89] which showed the emission quench at 525 nm by the addition of Au NPs to CdTe QDs via IFE and then visualized the enhancement of fluorescence in the presence of melamine. On the other hand, the emission at 620 nm was not affected with the existence of Au NPs, but the addition of melamine led to complete quenching of emission. Hence, this probe can be used as a dual-mode fluorescent probe for melamine detection as illustrated in Figure 13. The linear melamine concentration determined by this method was from 158.6 to 793 nM with a LOD of 87 nM. Moreover, melamine recoveries in milk samples were between 97.7~104.9% with 2.7~4.0% RSDs. Due to the dual "turn-on" and "turn-off" responses, the probe can be effective towards melamine assay in dairy samples.

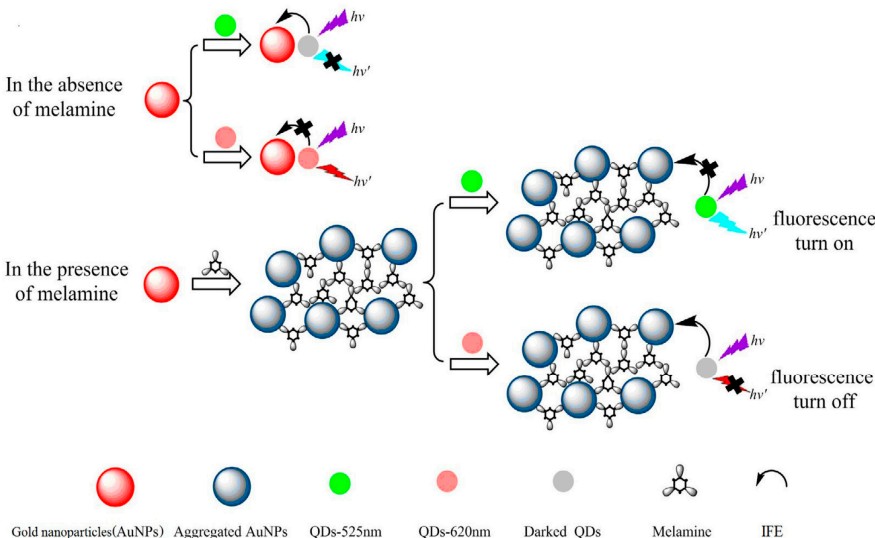

**Figure 13.** Schematic illustration of the dual-mode fluorescent probe for melamine detection (reproduced with the permission of Reference [89]).

A solid-state electrochemiluminescent sensor based on $C_{60}$/graphite-like carbon nitride nanosheet ($C_{60}$/g-$C_3N_4$ NS) hybrids was developed to quantify the melamine by Fu and co-workers [90]. The linear concentrations of melamine assay were between 0.5 to 27 pM and 27 pM to 19 nM with a LOD of 0.13 pM. This method can be considered as a suitable electrochemiluminescent (ECL) method for melamine discrimination due to its picomolar level detection ability. Moreover, the accuracy of this method is reliable to well established HPLC-MS analytical methods.

Similar to the above report, electrochemiluminescence property of the silica nanoparticles doped with $[Ru(bpy)_3]^{2+}$ and molecularly-imprinted polymer (MIP) were utilized to modify the glassy carbon electrode for selective sensing of melamine by Lian and co-workers [91]. Here, MIP was used as a recognition source towards melamine, which resulted in ECL signal enhancement of $[Ru(bpy)_3]^{2+}$. The linear regression was observed between 1 pM~100 nM with a LOD of $5 \times 10^{-13}$ mol/L. Moreover, the recoveries were from 90.0 to 104.0% with 4.2 to 5.3% RSDs in milk samples. Due to the lower

detection limits, this method can be used as an effective detection technique for melamine detection in real samples.

Metal-organic frameworks mediated nafion nanohybrid (MOFs@XC-72) composite was employed by Zhang et al. to modify the glassy carbon electrode towards the detection of melamine [92]. This hybrid system comprises of XC-72, MOFs-MIL-53 and Nafion which discriminates the melamine as represented in Figure 14. The linear correlation for melamine assay was between 0.04 to 10 µM with a LOD of 0.005 µM. In spiked milk samples, recoveries were from 98~103.5% with 3.1~4.8% RSDs. Due to the complication in electrode fabrication, this work can be only considered as an addition to electrode-based melamine sensors.

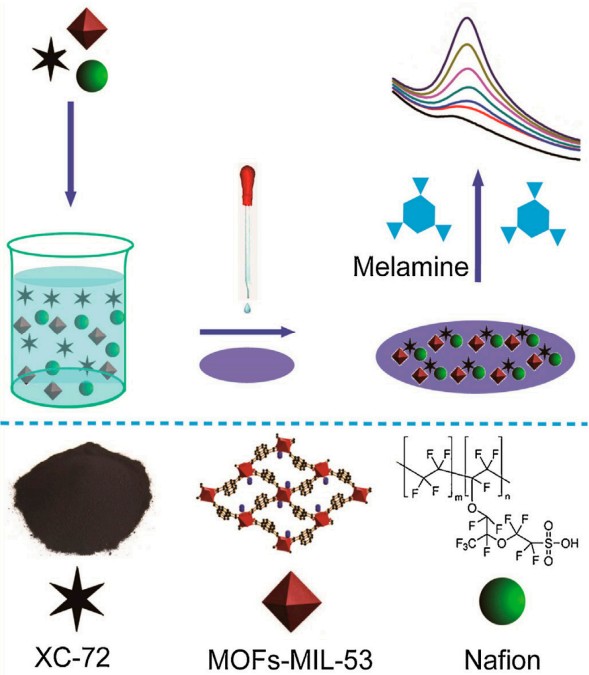

**Figure 14.** The illustration of the electrode preparation and application process (reproduced with permission from Reference [92]).

Gelatin-coated cerium oxide (Gel-CeO$_2$) nanospheres were reported in Jin's research work [93], in which he conveyed the peroxidase-like activity by means of H-bonding. The Gel-CeO$_2$ catalysed the oxidation of ABTS(2,20-azino-bis(3-ethylbenzothiazoline-6-sulfonic acid) diammonium salt) by H$_2$O$_2$ which resulted in the formation of blue color product. However, the above process was affected by the presence of melamine, which reacted with H$_2$O$_2$ via H-bonding and changed the color of the solution to pale. The detected concentration range of melamine was from 50 nM to 5.0 mM with a LOD of 5.5 nM. Melamine recoveries in spiked milk samples were from 98.6 to 104.0% with 4.2% RSD. Due the simplicity and the naked-eye detecting ability, this method can be used in real-time monitoring of melamine. Notably, all of the aforementioned composite-based research approaches enhance the contaminant detection such as melamine, clenbuterol, and other poisonous produces in food packaging industries [94].

## 5. Nanocrystals in Melamine Recognition

Nanocrystals are one of the modern candidates utilized in various analytical studies. For example, rare earth phosphate crystal decorated with Au NPs was reported by Chen et al. as a probe for the detection of biological aminothiols [95]. Mahalingam and co-workers presented the melamine sensing ability of 3,5-dinitrobenzoic acid (DNB)-capped up-conversion nanocrystals as described in Reference [96]. Wherein, DNB capped Er/Yb-NaYF$_4$ nanocrystals were investigated in melamine assay via quenching of emission as illustrated in Figure 15. The emission was restored upon the addition of

dilute acid. The detection limit by this method is remarkable and is estimated to be 2.5 nM. Hence it can be applied in real-time screening of melamine. Towards this exceptional efficient nanocrystals-based sensing approach, much contributions are required from other researchers.

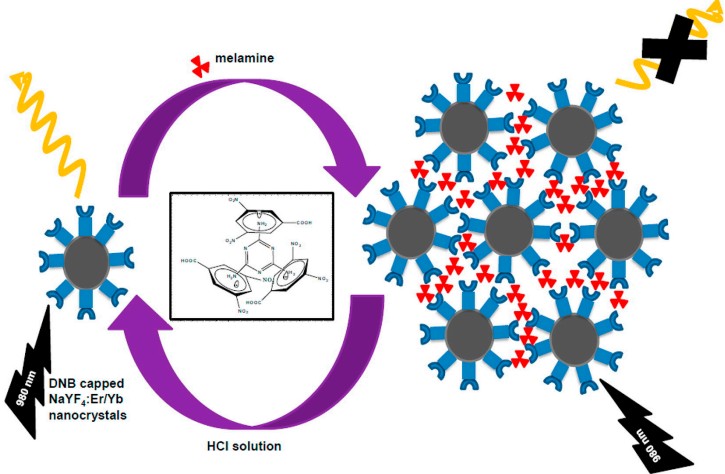

**Figure 15.** Scheme illustrating the interaction of melamine with 3,5-dinitrobenzoic acid (DNB) -capped NaYF$_4$:Er$^{3+}$/Yb$^{3+}$ nanocrystals (reproduced with permission from Reference [96]).

## 6. Nanoparticles in Melamine Quantification

Nanoparticles-based sensing approaches have become the modern research towards diverse analyte determination. Studies on the Ag NPs and Au NPs colorimetric sensors are particularly impressive [97]. For example, Cao's report described the Au NPs-based melamine assay as a suitable kit in various milk products [98]. They engaged the 5 nm Au NPs which were fabricated through sodium borohydride reduction and applied them towards the effective assay of melamine. Notably, this kit operated between the melamine concentrations of 7.93 μM to 0.95 mM with a LOD of 7.93 μM. The detection work can be completed within 10 min. However, various Au NPs-based rapid detection probes were also reported as described next. Li's group demonstrated the naked-eye visual detection process towards melamine recognition via peroxidase-like activity of bare Au NPs with the support of 3,3′,5,5′-tetramethlybenzidine (TMB)–H$_2$O$_2$ [99]. In this approach, the LOD of melamine by naked eye was established as 0.5 μM and the LOD was fixed to be 0.2 nM through standard deviation. The above method displays excellent recoveries and RSDs in spiked samples with the linearity ranged from 1 to 800 nM.

Chen et al. reported the bare gold nanoparticles as a probe for colorimetric discovery of melamine [100]. Similar to the above method, various label free or citrate-stabilized Au NPs with diverse sizes were reported in melamine discrimination [101–116]. Similarly, many capped or stabilized Au NPs were testified in melamine identification. Table 1 summarizes the detection methods, instruments employed, optimization status, time required for analyses, linear ranges, LODs, recoveries in spiked samples, and RSDs of those probes utilized so far in melamine analysis [98–166]. Information on instruments, optimization status, and time will determine the effectiveness and probable cost of the projected tactics, which provides helps for the future research. Surprisingly, most of these Au NPs-based probes detect the melamine adulteration via colorimetric responses. Only a few of them detect the melamine adulteration through SERS, fluorescence, light scattering, mass/ionization, peroxidase activity, and sonoluminescence techniques, etc. The majority of the label-free, unmodified or citrate-stabilized Au NPs detect melamine via aggregation of particles as shown in Figure 16. However, the LODs can be improved by changing the size of the Au NPs [105,115].

On the other end of the spectrum, Lu et al. presented the Au NPs-based fluorescent "turn-on" detection of melamine via mixing Au NPs with an organic probe [114]. Following this direction, diverse functional units stabilized Au NPs have been developed and exploited

in melamine discovery through colorimetric, SERS, FRET, CRET, U-Vis, light scattering, test strip, capillary electrophoresis, and fluorescence studies [98–166]. The mainstream of melamine sensing by functionalized Au NPs was attributed to H-bonding between stabilized Au NPs and melamine [117,118,120,124,126–130]. For example, Ai et al. developed the thiol containing cysteamine derivative "1-(2-mercaptoethyl)-1,3,5-triazinane-2,4,6-trione (MTT)" functionalized Au NPs which displayed the H-bonding mediated sensing of melamine [117] as illustrated in Figure 17. So far, many H-bond facilitated assays of melamine have been reported, which has led to saturation of this research strategy. Therefore, to further extend the Au NPs-based on-site melamine assay, collaboration from other research fields—such as opto-electronic techniques—is required.

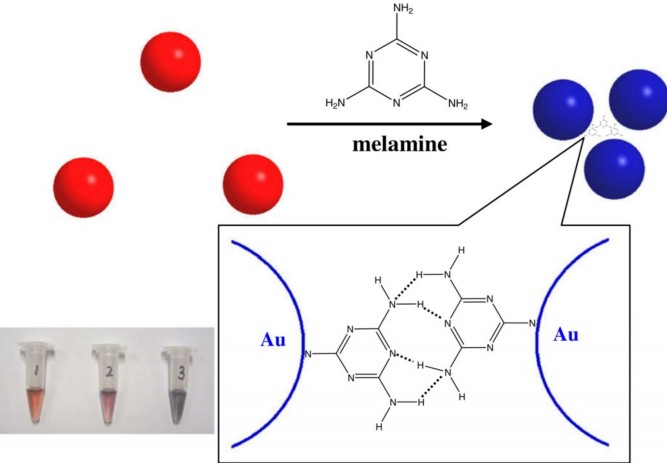

**Figure 16.** Schematic representation of the GNPs colorimetric mechanism for melamine detection. The insert is photographs of solution of (1) 400 μL GNPs + 20 μL $H_2O$, (2) 400 μL GNPs + 20 μL melamine ($5 \times 10^{-3}$ g/L), and (3) 400 μL GNPs + 20 μL melamine ($20 \times 10^{-3}$ g/L). Experimental condition: GNPs, 1.4 μM; incubation time, 1 min; reaction temperature, room temperature (~20 °C).

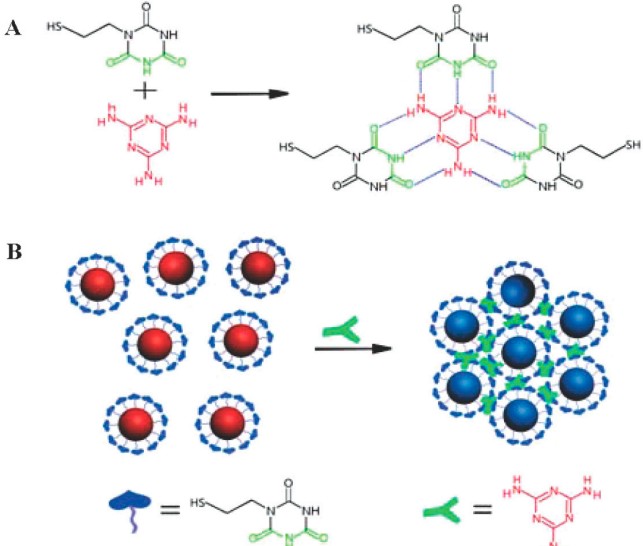

**Figure 17.** (**A**) Hydrogen-bonding recognition between melamine and cyanuric acid derivative. (**B**) Colorimetric detection of melamine using the 1-(2-mercaptoethyl)-1,3,5-triazinane-2,4,6-trione (MTT)-stabilized gold nanoparticles (reproduced with permission from Reference [117]).

**Table 1.** Summary of methods, instruments, optimization status, time, linear ranges, detection limits (LODs), recoveries and relative standard deviations (RSDs) of Au NPs-based probes in melamine detection.

| Au NPs-Based Probe | Method of Detection | Instruments Employed | Optimization Status and Time Required | Linear Range | LOD | Recoveries of Spiked Samples | RSDs | Ref. |
|---|---|---|---|---|---|---|---|---|
| Bare Au NPs | Colorimetric | Naked eyes and UV-Vis spectrometer | Complicated and 10 min | 7.93 μM–0.95 mM | 7.93 μM | NA | NA | [98] |
| Bare Au NPs | Colorimetric (peroxidase-like activity) | Naked eyes and UV-Vis spectrometer | Complicated and 30 min | 1–800 nM | 0.2 nM | 94.55–120.50% | 0.07–0.99% | [99] |
| Bare Au NPs | Colorimetric | Naked eyes and UV-Vis spectrometer | Moderate and 7 min | 39.64 nM–1.59 μM | 1.59 nM | 97.6–107% | 0.8–2.4% | [100] |
| Citrate-stabilized Au NPs | Colorimetric | Naked eyes and UV-Vis spectrometer | Moderate and 20 min | 0–634 μM | 19.8 μM | NA | NA | [101] |
| Citrate-stabilized Au NPs | Colorimetric | Naked eyes and UV-Vis spectrometer | Moderate and 2 min | 0–1.9 μM | 198 nM | NA | NA | [102] |
| Label free Au NPs | Colorimetric | Naked eyes and UV-Vis spectrometer | Moderate and 12 min | 1.59–79.3 μM | 3.2 μM | 97–105% | 0–2% | [103] |
| Citrate-stabilized Au NPs | Fluorescent | UV-Vis and PL spectrometer | Moderate and NA | 0.8–80 nM | 0.61 nM | 97.92–98.54% | NA | [104] |
| Citrate-stabilized Au NPs | Colorimetric | Naked eyes and UV-Vis spectrometer | Moderate and 1 min | NA | 39.64 nM | NA | NA | [105] |
| Citrate-stabilized Au NPs | Colorimetric | Naked eyes and UV-Vis spectrometer | Moderate and 10 min | 0.79–15.9 μM | 0.4 μM | 95–105% | 1.28–10.53% | [106] |
| Citrate-stabilized Au NPs | Colorimetric | Naked eyes and UV-Vis spectrometer | Moderate and 15 min | 1.6–159 μM | 15.86 μM | 105–116% | NA | [107] |
| Citrate-stabilized Au NPs | Colorimetric | Naked eyes, DLS and UV-Vis spectrometer | Complicated and NA | 1–100 μM | 33 nM, 23.7 nM and 89 nM | 91–104% | 0.23–4.43% | [108] |
| Unmodified Au NPs | Colorimetric | Naked eyes and UV-Vis spectrometer | Moderate and >15 min | 0.95–3.9 μM | 317 nM | NA | NA | [110] |
| Unmodified Au NPs | Colorimetric | Naked eyes and UV-Vis spectrometer | Moderate and NA | 0.79–79 μM | >3.2 μM | NA | NA | [111] |
| Unmodified Au NPs | Colorimetric | Naked eyes and UV-Vis spectrometer | Moderate and 20 min | 0–2 μM | 555 nM | 90–120 % | NA | [112] |
| Unmodified Au NPs | Colorimetric | Naked eyes and UV-Vis spectrometer | Moderate and NA | 0.198–2.4 μM & 0.792–7.13 μM | 182 nM and 729 nM | 97.5–101.1% | NA | [113] |
| Citrate-stabilized Au NPs | Fluorescent | PL spectrometer | Moderate and >30 min | 10 nM–4 μM | 3 nM | 92–108% | 0.80–4.21% | [114] |
| Citrate-stabilized-Au NPs | Fluorescent | DLS and PL spectrometer | Complicated and 5 min | 40–700 nM | 0.35 nM | 97–100% | 2.1–4.28% | [115] |
| Citrate and DNA-Au NPs | Colorimetric | Naked eyes and UV-Vis spectrometer | Moderate and 5 min | NA | 41.7 nM amd 46.5 nM | 82.9–102.6% | 0.80–2.06% | [116] |

**Table 1.** *Cont.*

| Au NPs-Based Probe | Method of Detection | Instruments Employed | Optimization Status and Time Required | Linear Range | LOD | Recoveries of Spiked Samples | RSDs | Ref. |
|---|---|---|---|---|---|---|---|---|
| 1-(2-mercaptoethyl)-1,3,5-triazinane-2,4,6-trione (**MTT**)-stabilizedAu NPs | Colorimetric | Naked eyes and UV-Vis spectrometer | Mild and 5 min | 7.93–39.64 μM | 19.82 nM | NA | NA | [117] |
| Hexadecy ltrimethyl ammonium chloride (**CTAC**)-stabilized Au NPs | Colorimetric | Naked eyes and UV-Vis spectrometer | Moderate and 50 min | 1 nM–10 μM | 0.8 nM | NA | NA | [118] |
| 11-Mercapto-undecanoic acid (**MUA**)-stabilized Au NPs | Capillary electrophoresis-UV | Naked eyes and UV absorbance detector | Complicated and 90 min | 1–1000 nM | 77 pM | 97–101% and 95–99% | NA | [119] |
| Polythymine (**Poly T$_n$**)-stabilized Au NPs | Colorimetric | Naked eyes, DLS and UV-Vis spectrometer | Moderate and 30 min | 80–1000 nM | 20 nM | NA | NA | [120] |
| 18-crown-6Ether-functionalized Au NPs | Colorimetric | Naked eyes and UV-Vis spectrometer | Moderate and >1 min | 79.3 nM–3.96 μM | 47.57 nM | 98.4–105.6 % | 1.7–5.8% | [121] |
| Cysteamine-modified Au NPs | Colorimetric | Naked eyes and UV-Vis spectrometer | Moderate and >30 min | 7.92 μM–1.59 mM | 7.92 μM | NA | NA | [122] |
| Citrate-stabilized Au NPs with Fluorescein | FRET | PL spectrometer | Moderate and >12 min | 0.1 μM–4 μM | 1 nM | NA | NA | [123] |
| 3-mercapto-1-propane-sulfonate-modifiedAu NPs | Colorimetric | Naked eyes and UV-Vis spectrometer | Moderate and 30 min | 10–150 nM & 150–600 nM | 8 nM | 98.0–104.5% 93.6–101.6% | 1.6–3.7% and 3.9–5.9% | [124] |
| 4-mercaptopyridine-modified Au NPs | SERS | UV-Vis and Raman spectrometer | Moderate and >0.5 min | 3.96–793 nM | 793 pM | 88.5–119.2% | NA | [125] |
| pyrocatechol-3,5-disodiumsulfonate-stabilized Au NPs | Colorimetric | Naked eyes and UV-Vis spectrometer | Moderate and 80 min | 4.8 nM–1.6 μM | 0.64 nM | 93–107 % | NA | [126] |
| ssDNA-stabilized Au NPs | Resonance Rayleigh Scattering (RRS) and Cat RRS | Eclipse fluorescence spectro-photometer | Moderate and >30 min | 15–650 nM and 5–38 pM | 7.8 and 3 pM | 99.2–100% | 0.8–1.7% | [128] |
| Thioglycolic-Acid-Modified Au NPs | Colorimetric | Naked eyes and UV-Vis spectrometer | Moderate and >15 min | 0–19.66 μM | NA | 101.1–102% | 1.6–2.3% | [129] |

**Table 1.** *Cont.*

| Au NPs-Based Probe | Method of Detection | Instruments Employed | Optimization Status and Time Required | Linear Range | LOD | Recoveries of Spiked Samples | RSDs | Ref. |
|---|---|---|---|---|---|---|---|---|
| 2,4,6-trinitrobenzene-sulfonic acid (TNBS) tailored Au NPs | Colorimetric | Naked eyes and UV-Vis spectrometer | Moderate and >10 min | 0–634 nM | 39.64 nM | NA | NA | [130] |
| Citrate-stabilized Au NPs | Strip method | immuno-chromatographic strip analyzer | Complicated and NA | 23.8–99 nM | 35.4 nM | NA | NA | [131] |
| Pyridine-3-Boronic Acid-modified Au NPs | Colorimetric | Naked eyes and UV-Vis spectrometer | Moderate and >20 min | 60 nM–1.6 μM | 30 nM | 95–102% | NA | [132] |
| Bare Au NPs | SERS | Raman spectrometer | Complicated and >15 min | 1.6–159 μM | 3.1 μM | 95–109% | 0.77–4.21% | [133] |
| 3-amino-5-mercapto-1,2,4-triazole-capped Au NPs | Colorimetric & Fluorimetry | Naked eyes, UV-Vis and PL spectrometer | Moderate and NA | 0.1–1 nM | 10 fM | NA | NA | [134] |
| ssDNA-modified Au NPs | Colorimetric | Naked eyes and UV-Vis spectrometer | Moderate and 40 min | 0.1–1.0 μM | 34 nM | 94–120% | NA | [135] |
| Cysteamine-modified Au NPsdified | Colorimetric | Naked eyes and UV-Vis spectrometer | Moderate and 45 min | 0.08–1.6 μM | 80 nM | 98–102% | 1.7–2.6% | [136] |
| Au NPs synthesized by ellagic acid (EA) | Colorimetric | Naked eyes and UV-Vis spectrometer | Mild and 30 min | 16 nM–160 μM | 1.6 nM | 93–106% | NA | [137] |
| Aptamer-modified Au NPs | Colorimetric | Naked eyes and UV-Vis spectrometer | Moderate and 30 min | 1.2–2.4 μM and 2.4–20.62 μM | 793 nM | 95–105 % | 3.9% | [138] |
| Citrate-stabilized Au NPs | Chemiluminescence resonance energy transfer (CRET) | Chemiluminescence analyzer, PL and UV-Vis spectrometer | Mild and 45 min | 3.2 pM–0.32 μM | 0.3 pM | 94.1–104.2% | 1.5–4.5% | [139] |
| p-DNA-modified Au NPs | Colorimetric and Dynamic Light Scattering (DLS) | Naked eyes, DLS and UV-Vis spectrometer | Moderate and >3 min | 39.64 nM–2.54 μM | 15.9 nM | NA | NA | [140] |
| Citrate and dodecasodium salt of phytic acid functionalized Au NPs | SERS | Raman spectrometer | Moderate & 90 min | 10–100 μM | 5 μM | 93.6% | NA | [141] |
| 3-Mercapto-propionic acid functionalized Au NPs | Colorimetric | Naked eyes and UV-Vis spectrometer | Moderate & 10 min | 4.8–333 nM | 3.2 nM | 96–105% | NA | [142] |
| Citrate stabilized Au NPs with Rhodamine B | FRET | PL and UV-Vis spectrometer | Moderate & >40 min | 39.64 nM–7.93 μM | 1.43 nM | 95.9–102.2% | 0.8–3.0% | [143] |
| Acetylated chitosan-stabilized Au NPs | Colorimetric | Naked eyes, CV and UV-Vis spectrometer | Moderate & NA | 396 nM–7.93 μM | 389 nM | 94–111% | NA | [144] |

**Table 1.** *Cont.*

| Au NPs-Based Probe | Method of Detection | Instruments Employed | Optimization Status and Time Required | Linear Range | LOD | Recoveries of Spiked Samples | RSDs | Ref. |
|---|---|---|---|---|---|---|---|---|
| 1,4-dithiothreitol-modified (**DTT**) Au NPs | Colorimetric | Naked eyes and UV-Vis spectrometer | Moderate and 5 min | 80 nM–1.5 μM | 24 nM | 96–103% | NA | [145] |
| Au NPs synthesized by Methanobactin (Mb) | Colorimetric | Naked eyes and UV-Vis spectrometer | Moderate and 50 min | 0.39–3.97 μM | 0.238 μM | 97.5–103.1% | 0.8% | [146] |
| Thymine derivative-functionalized Au NPs | Colorimetric | Naked eyes and UV-Vis spectrometer | Moderate and 10 min | 0.75–5.00 μM | 3.5 nM | 96.5–102.0% | 4.0–11.8% | [147] |
| $H_2O_2$–Au NPs | Colorimetric | Naked eyes and UV-Vis spectrometer | Moderate and 35 min | 0.4–160 μM | 0.078 μM | 90–113.7% | NA | [148] |
| Up-conversion nanoparticles (UCNPs) and Au NPs | FRET | PL spectrometer | Moderate and 12 min | 32–500 nM | 18 nM | 98.8–102% | 2.32–4.44% | [149] |
| Citrate-stabilized Au NPs | Fluorescent and UV-Vis | UV-Vis and PL spectrometer | Complicated and 60 min | 0.4–2 μM | 0.88 μM | NA | NA | [150] |
| Polythymine (T) aptamer-modified Au NPs | SERS | Raman spectrometer | Complicated and NA | 0–31.7 fM | 7.9 fM | 97.3–109.53% | NA | [151] |
| *p*-chlorobenzenesulfonic acid-modified Au NPs | Colorimetric | Naked eyes and UV-Vis spectrometer | Moderate and 15 min | 0.6–1.5 μM | 2.3 nM | 97.9–103% | 0.1–6.5% | [152] |
| BSA conjugated Au NPs | Colorimetric | Signal amplified lateral flow strip | Complicated and NA | 7.93 nM–1.59 μM | 11.1 nM | NA | NA | [153] |
| Unmodified Au NPs | Colorimetric | Naked eyes and UV-Vis spectrometer | Moderate and 30 min | 0–0.9 μM | 33 nM | 99.2–111% | 0.56–1.91% | [154] |
| Cysteamine-stabilized Au NPs | Colorimetric | Naked eyes and UV-Vis spectrometer | Moderate and 6 min | 1–24 nM | 0.389 nM | 92.8–112.2% | NA | [155] |
| amine-ended dual thiol ligand functionalized Au NPs | Colorimetric | Naked eyes and UV-Vis spectrometer | Complicated and NA | NA | NA | NA | NA | [156] |
| Cellulose-coated Au NPs | SERS | Raman spectrometer | Moderate and 15 min | 0–79.3 μM | 7.93 μM | 87.6–92.3% | NA | [157] |
| Triton X-100-modified Au NPs | Colorimetric | Naked eyes and UV-Vis spectrometer | Moderate and NA | 0.75–1.75 μM | 5.1 nM | 99–111% | 0.73–2.91% | [158] |
| uracil 5′-triphosphate sodium-modified Au NPs | Colorimetric and light scattering | Naked eyes, DLS and UV-Vis spectrometer | Moderate and 30 min | 300–900 nM and 200–950 nM | NA | 98.5–104% | 3.6–4.6% | [159] |

**Table 1.** *Cont.*

| Au NPs-Based Probe | Method of Detection | Instruments Employed | Optimization Status and Time Required | Linear Range | LOD | Recoveries of Spiked Samples | RSDs | Ref. |
|---|---|---|---|---|---|---|---|---|
| Citrate-stabilized Au NPs | SERS | UV-Vis and Raman spectrometer | Moderate and 10 min | 0–1.59 µM | 793 nM | NA | NA | [160] |
| Unmodified Au NPs | SERS | Raman spectrometer | NA | 0–79.3 µM | 793 nM | NA | NA | [161] |
| Citrate stabilized Au NPs | SERS | Raman spectrometer | Moderate and >30 min | 2.5–39.64 µM | 1.35 µM | 96.3–99.9% | 3.8–9.6% | [162] |
| Citrate stabilized Au NPs | Colorimetric & SERS | Naked eyes and Raman spectrometer | Moderate and 20 min | 0–1.98 µM | NA | NA | NA | [163] |
| Citrate stabilized Au NPs | Mass Analysis | surface-assisted laser desorption/ionization mass spectrometer | Complicated and >0.5 min | NA | NA | NA | NA | [164] |
| Citrate stabilized Au NPs | Sonoluminescence | Sonoluminescence analyzer | Moderate and >12 min | 10–240 nM | 3 nM | 95% | NA | [165] |
| SiO$_2$ shell-isolated Au NPs | SERS | UV-Vis and Raman spectrometer | Moderate and >6 min | 3.96–39.64 µM | 7.93 µM | 94.6–102.5% | 5.4–9.5% | [166] |

NA = Not available; mM = millimole; µM = micromole; nM = nanomole; pM = picomole; fM = femtomole; min = minutes.

Chang and co-workers presented the assay of melamine through UV-Vis followed by capillary electrophoresis [119]. They demonstrated the 11-mercaptoundecanoic acid-capped gold nanoparticles (MUA–AuNPs) which recognized melamine via aggregation of Au NPs. The dithiothreitol (DTT) was used to extract the melamine from the supernatant through capillary electrophoresis. This method can be validated as a suitable detection and extraction technique for melamine present in dairy products and food products. However, optimization of the above technique is most likely to be complicated, and hence real-time reliability is still in question. Likewise, a complex, which performs melamine recognition, is realized by 18-crown-6-ether-functionalized Au NPs and methanobactin mediated Au NPs synthesize [121,146]. The effective melamine detection is demonstrated by 18-crown-6-ether-functionalized Au NPs via complex formation as illustrated in Figure 18.

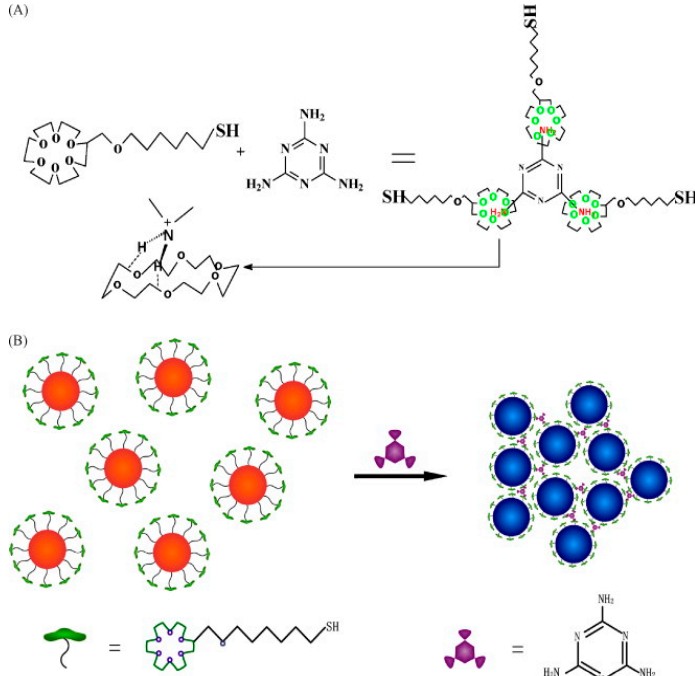

**Figure 18.** Schematics of the melamine sensing with 18-crown-6-thiol-modified GNPs. (**A**) Specific interaction in the complex between the crown ether and melamine; (**B**) melamine-induced aggregation of 18-crown-6-thiol-modified GNPs (reproduced with permission from Reference [121]).

Aggregation and electrostatic forces involved tactics were also described by a few reports [122,134,144,154,158]. However, these methodologies still require more attention. On the other hand, a few researchers have attempted to utilize the Au NPs-based probes towards melamine quantification by fluorescence resonance energy transfer (FRET) and chemiluminescence resonance energy transfer (CRET), which may possibly demonstrate the effective assay of melamine in the near future [123,139,143,149]. Besides, SERS-based detection of melamine has also been authenticated as a more efficient tactic. Many researchers reported the functionalized or label free or bare Au NPs towards the discrimination of melamine through SERS studies [125,133,141,151,157,160–163,166]. Many of them revealed great applicability with exceptional LODs. Hence such Au NPs probes for SERS-based melamine identification in milk stuffs is highly appreciated.

In this track, melamine recognition was demonstrated by Wang and co-workers through SERS spectra and in filter paper strip [141]. Tri-sodium citrate and dodecasodium of phytic acid (IP6) dual-functionalized Au NPs (IP$_6$-TC@Au NPs) were applied in the screening process. The above tactic permits the on-site broadcast of melamine in food products. In a similar fashion, Dong et al. developed an aptamer-modified SERS nanosensor and oligonucleotide chip to quantify the melamine via multi-hydrogen-bond formation between thymine and melamine [151]. This method showed a

LOD at picogram level, and hence such design is highly desirable. The BSA conjugated Au NPs were utilized by Sun and collaborators [131]. Therein, they evaluated the effectiveness of their method in 50 raw milk samples and the results agreed well with chromatographic and mass data. Hence, such strip method can be authenticated as a rapid screening procedure for real-time monitoring of melamine.

BSA-Au NPs-based lateral flow immunoassay was proposed by Zhong and co-workers For rapid discrimination of melamine [153]. This method was also legalised as an effective procedure for melamine detection, but still much focus is required to improve its applicability and detection limits. Subsequently, the catalytic property of antibody conjugated Au NPs was exploited towards melamine determination by Knopp's group [150]. Surface-assisted laser desorption/ionization mass spectrometry (SALDI-MS) was engaged in the quantification of melamine in infant formula and grain powder by Hsieh and co-workers [164]. The melamine (MEL), ammeline (AMN), and ammelide (AMD) were determined by SALDI-MS using Au NPs. This SALDI-MS quantified MEL, AMN, and AMD via mass peaks at m/z 127.07, 128.05, and 129.04 were correlated to $[MEL + H]^+$, $[AMN+H]^+$, and $[AMD + H]^+$ ions. Moreover, LODs of MEL, AMN, and AMD were estimated as 5, 10, and 300 nM, respectively. Hence, the above tactics of combining SALDI-MS with Au NPs can be validated for rapid screening of melamine. Next, a sonoluminescence-based approach was proposed for melamine discrimination by Liu and collaborators [165], in which the engaged Au NPs were aggregated in the presence of melamine. Therefore, it belongs to the category of the aggregation-induced sensors. Moreover, such sonoluminescence-based analyte sensing still requires more research for on-site inspection of melamine.

Similar to the Au NPs, the Ag NPs mediated discovery of melamine in dairy milk and food products are also well established in modern science [167]. These Ag NPs facilitated melamine assays were operated through colorimetric, SERS, resonance scattering, and fluorescence responses [168–189]. However, the main working principles of Ag NPs assisted melamine sensors were mainly by means of colorimetric responses, initialized through H-bonding, self-assemblies, and electrostatic forces supported aggregation of nanoparticles [16–173,176–178,180–182,185–187]. An example of electrostatic forces induced nanoparticles aggregation by melamine is illustrated in Figure 19. Kumar et al. demonstrated the unmodified Ag NPs towards the detection of melamine via colorimetric response from yellow to red [170]. On the other hand, Ag NPs were also employed in SERS-based quantification of melamine [175,179,184,188,189]. Table 2 summarizes the detection methods, instruments employed, optimization status, time required for analyses, linear ranges, LODs, recoveries in spiked samples, and RSDs of Ag NPs-based probes exploited in melamine recognition.

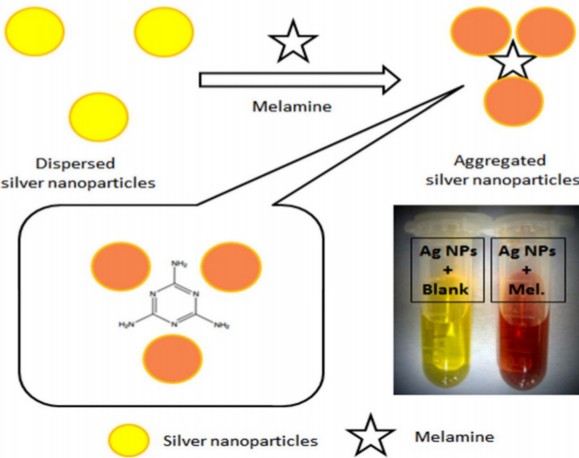

**Figure 19.** Schematic illustration of melamine detection and visual color change of Ag NPs after addition of melamine. The insert is a photograph of visual color change of Ag NPs upon addition of 1 mg/L of melamine (reproduced with permission from Reference [170]).

**Table 2.** Summary of methods, instruments, optimization status, time, linear ranges, detection limits (LODs), recoveries and relative standard deviations (RSDs) of Ag NPs-based probes in melamine detection.

| Ag NPs-Based Probe | Method of Detection | Instruments Employed | Optimization Status and Time Required | Linear Range | LOD | Recoveries of Spiked Samples | RSDs | Ref. |
|---|---|---|---|---|---|---|---|---|
| Label-free Ag NPs | Colorimetric | Naked eyes and UV-Vis spectrometer | Moderate and 50 min | 2–250 μM | 2.32 μM | 88.83–114. 29% | 2.04–3.10% | [168] |
| Bare Ag NPs | Colorimetric | Naked eyes and UV-Vis spectrometer | Moderate and30 min | 40–880 nM | 10 nM | 97.5–105% | 2.98–4.83% | [169] |
| Unmodified Ag NPs | Colorimetric | Naked eyes and UV-Vis spectrometer | Moderate and 20 min | 0–15.86 μM | 0.32 pM | 92.5–99.4% | 5.26–8.18% | [170] |
| Bare Ag NPs | Colorimetric | Naked eyes and UV-Vis spectrometer | Moderate and 30 min | 0.26 pM–11.89 μM | 71.4 nM | 61.9–96.3% | NA | [171] |
| *p*-nitroaniline-modified Ag NPs | Colorimetric | Naked eyes and UV-Vis spectrometer | Moderate and 10 min | 0.79 μM–79.3 mM | 0.79 μM | NA | NA | [172] |
| Dopamine-stabilized Ag NPs | Colorimetric | Naked eyes and UV-Vis spectrometer | Moderate and 60 min | 0.08–10.0 μM | 79.3 nM | 92–105% | NA | [173] |
| ssDNA-stabilizedAg NPs | Resonance scattering | PL and CD spectrometers etc . . . | Complicated and 90 min | 0.05–3 μM | 23.8 nM | 98.7–100.9% | 0.8–3.6% | [174] |
| Oleylamine capped Ag NPs | SERS | Raman spectrometer | Complicated and >0.5 min | 0.1–100 μM | 100 nM | NA | NA | [175] |
| *β*-cyclodextrin-functionalized Ag NPs | Colorimetric | Naked eyes and UV-Vis spectrometer | Moderate and 30 min | 1 mM–50 μM | 4.98 μM | 80.5–109.02% | 2.27–3.03% | [176] |
| Chromotropic acid (CTA)-modified Ag NPs | Colorimetric | Naked eyes and UV-Vis spectrometer | Moderate and 6 min | 0.10–1.5 μM | 36 nM | 91–105 % | NA | [177] |
| Sulfanilic acid-modified Ag NPs | Colorimetric | Naked eyes and UV-Vis spectrometer | Moderate and 5 min | 0.1–3.1 μM | 10.6 nM | 97–109% | 0.9–1.9% | [178] |
| Starch-coated Ag NPs | SERS | UV-Vis and Raman spectrometers | Complicated and NA | 15.9 μM–0.4 mM | 4.8 nM | 94–104% | 2.39–4.53% | [179] |
| Bio-functionalized Ag NPs | Colorimetric | Naked eyes and UV-Vis spectrometer | Complicated and 20 min | 0.015–1 mM | 2 μM | 96–122% | 0.44–2.22% | [180] |
| Sodium D-gluconate-stabilized Ag NPs | Colorimetric | Naked eyes and UV-Vis spectrometer | Moderate and NA | 0.5–500 μM | 476 nM | 90–98% | NA | [181] |
| Polyelectrolyte-stabilized Ag NPs | Colorimetric and Fluorescence | Naked eyes, UV-Vis and PL spectrometers | Complicated and 20 min | 1 nM–1.5 μM and 1.5 nM–150 μM | 0.1 and 0.45 nM | 99–114% | 1.66–4.37% | [182] |

**Table 2.** *Cont.*

| Ag NPs-Based Probe | Method of Detection | Instruments Employed | Optimization Status and Time Required | Linear Range | LOD | Recoveries of Spiked Samples | RSDs | Ref. |
|---|---|---|---|---|---|---|---|---|
| Citrate and Borohydride stabilized Ag NPs | Colorimetric | Naked eyes and UV-Vis spectrometer | Moderate and NA | NA | NA | NA | NA | [183] |
| Ag NPs monolayer film | SERS | Raman spectrometer | Complicated and >15 min | 0.79 pM–39.6 μM | 0.32 pM | 90–95.4% | 3.7–6.9% | [184] |
| Tannic acid-stabilized Ag NPs | Colorimetric | Naked eyes and UV-Vis spectrometer | Moderate and 20 min | 0.05–1.4 μM | 0.01 μM | 98.5–106.5% | 1.04–3.19% | [185] |
| Bio-functionalized Ag NPs | Colorimetric | UV-Vis and Raman spectrometers | Moderate and >1 min | 0.79–40 μM | 0.79 and 3.96 μM | 96% | NA | [186] |
| Green synthesized Ag NPs | Colorimetric | Naked eyes and UV-Vis spectrometer | Moderate and NA | 0.79–79.3 μM | 793 nM | NA | NA | [187] |
| Acid-directed synthesis of Ag NPs | SERS | Raman spectrometer | Complicated and NA | 0–0.396 mM | 39.64 μM | NA | NA | [188] |
| Chitosan-modified Ag NPs | SERS | Chromatography and Raman spectrometer | Complicated and >1 min | 0–79.3 μM | 7.93 μM | NA | NA | [189] |

NA = Not available; mM = millimole; μM = micromole; nM = nanomole; min = minutes.

Next, as displayed in Figure 20, the chromotropic acid (CTA)-capped AgNPs towards H-bonding facilitated the sensing of melamine [177]. The –NH$_2$ group of melamine H-bonded with –SO$_3$ group of chromotropic acid. Hence, such functionalized Ag NPs can be applied towards the determination of melamine and other analytes. Zhu et al. presented the polyelectrolyte functionalized Ag NPs for selective assay of melamine through colorimetric and fluorescence responses [182], in which the aggregation of particles led to sensor responses in the presence of melamine. The values of effective recovery, linear ranges, and LODs (see Table 2) demonstrated the suitability for real-time screening of melamine in food stuffs. Therefore, designing such probes with dual responses are much desired. Identification of melamine in dairy products was meritoriously carried out by Ag NPs via SERS and resonance scattering studies. As shown in Table 2, the Ag NPs-based SERS sensors have competitive linear ranges, recoveries, and LODs. Therefore, such designs are much anticipated for real-time examination of melamine. A resonance scattering-based detection procedure was described by Liang and co-workers [174]. The linear recovery ranges and LOD reported by this technique were found to be decent, and hence can be engaged in future analytical practicalities.

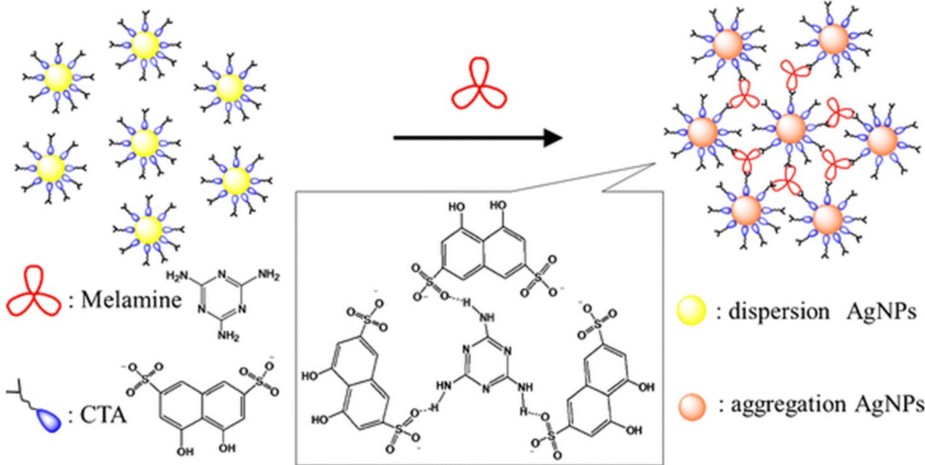

**Figure 20.** Schematic illustration of possible mechanism for sensing melamine based on CTA-AgNPs (reproduced with permission from Reference [177]).

Other than the Au NPs and the Ag NPs, Au-Ag bi-metallic nanoparticles (CSP NPs) were also employed in the sensing of melamine. For example, Li et al. described the bifunctional chitosan-modified popcorn-like Au-Ag nanoparticles for the colorimetric and SERS-based determination of melamine in milk powder samples [190]. The colorimetric response of CSP NPs were attributed to the peroxidase-like catalytic activity (in presence of melamine) in the oxidation of 3,3′,5,5′-tetramethylbenzidine (TMB) by H$_2$O$_2$. Wherein, the linear range of detection was from 10 nM to 50 μM with a LOD of 8.51 nM. Moreover, the recoveries were between 90.86~106.29% with 1.66~4.29% RSDs. Therefore, such bi-metallic NPs can be utilized towards real-time examination of melamine in dairy products. On this path, sodium polystyrene sulfonate capped Cu$_{2-x}$Se nanoparticles (Cu$_{2-x}$Se@PSS) were reported for melamine recognition by means of aggregation-induced superior peroxidase-like activity [191]. Huang's research unit explained the peroxidase-like activity of Cu$_{2-x}$Se@PSS in the presence of melamine, which helped the oxidation of 3,3′,5,5′-tetramethylbenzidine (TMB-colorless) by H$_2$O$_2$ to blue-colored oxidized TMB. The NPs displayed the linearity for melamine with range from 4.7 nM to 29.7 mM and a LOD of 1.2 nM. However, the applicability of this work still requires further proof in real samples.

Se NPs were proposed by Wang and collaborators through test strips analysis towards the sensing of melamine [192]. This method shows a LOD of 1 Ag/Kg in liquid milk, which confirms its effectiveness in melamine discrimination. Shen et al. demonstrated the assay of melamine by dopamine conjugated methoxypoly (ethylene glycol) carboxyl acid (mPEG-COOH) and

(2,4,6-trioxo-1,3,5-triazin-1-yl) acetic acid-functionalized Fe/Fe$_3$O$_4$ nanoparticles (TTAA–Fe/Fe$_3$O$_4$ NPs) [193]. The melamine recognition was attributed to the H-bonding interaction between functional moieties with melamine. The probe displayed the linearity from 0 to 16 μM with a LOD of 2 μM. This work does establish its applicability on real samples, and hence can contribute to melamine assay process. Li and co-workers described the determination of melamine by means of C-dots stabilized Ag NPs [194], in which the C-dots acted both as a reducer and a stabilizer. The above work demonstrated the melamine detection via visible and fluorescence responses between 0~2 μM and 2~20 μM, respectively, with a LOD of 30 nM. Moreover, recoveries of melamine in raw milk samples were between 95.71~113.58% with 2.9~4.3% RSDs. Other than the detection of melamine by functionalized NPs, melamine-stabilized NPs were also used in the assay of food contaminants [195]. Therefore, such a scheme can be further directed towards the identification of melamine in real samples as well [10].

## 7. Nanorods and Nanotubes in Melamine Assay

Due to the requirement of diverse techniques for melamine assay in food products, researchers also explored the possibility of using the nanorods and nanotubes for melamine discrimination, as discussed in this section. In 2010, Wei's group reported the melamine sensing through bis(8-quinolinolato)zinc(II) complex nanorod arrays [196]. In which, nanorods with diameters of 250 to 320 nm and lengths of ~25 μm were fabricated via liquid–liquid interfacial precipitation in the pores of porous anodic aluminum oxide membrane. These arrays displayed the linearity to melamine from 39.6 to 238 nM, and hence can be used as a suitable method for melamine assay. However, much work is still needed for this type of array. Following the above work, Chen and co-workers detected the melamine using the SERS ability of ZnO/Au nanoneedle arrays [197]. This work demonstrated the melamine determination from 10 nM to 100 μM with a LOD of 10 nM. Moreover, the authors also confirmed the melamine quantifying ability of ZnO/Au nanoneedles in egg-white solution. Due to the complications in optimization conditions, this method requires further modification for on-site melamine screening. In a similar fashion, Ag-nanoparticle-modified single Ag nanowires (Ag NP/Ag NWs) were exploited in the SERS-based quantification of melamine [198]. The Ag NWs were synthesized by solvothermal method and then decorated with the Ag NPs. The Ag NP/Ag NW showed great SERS response to melamine concentrations from 10 nM to 22 μM with a LOD of 10 n M. Moreover, this rapid detection approach was validated by quantifying melamine in milk solution with the detected melamine concentration as low as 50 n M. Even though the technique attested is one of the best tactics, complications involved in the optimization procedures still need to be rectified.

SERS effect engaged with Ag NPs coated ZnO (Ag@ZnO) nanorod arrays for the discovery of melamine was reported by Xu and collaborators [199]. However, much focus is still required to establish the melamine quantification by Ag@ZnO nanorod arrays. On the other hand, carbon nanotubes-based electrochemical assay of melamine was described by Li and Zhao research groups [200,201]. Li et al. employed the nanocomposite of hydroxyapatite/carbon nanotubes for the determination of melamine using Ascorbic acid (AA) as a recognition element [200]. Under the optimum condition, the decrease in anodic peak current of AA was linearly proportional to the melamine concentrations from 10 to 350 nM with a LOD of 1.5 nM. This electrochemical method demonstrated good recoveries in infant formula and milk samples, which were between 98.5~102.5% with 1.32~2.58% RSDs. On a similar track, Zhao et al. presented the assay of melamine by employing the glassy carbon electrode coated with a multi-wall carbon nanotube/chitosan composite [201]. The linear melamine concentrations were from 9.9 to 190 nM with a LOD of 3 nM. Moreover, this work was also demonstrated in milk samples with a recovery rate of 104.8%. These electrochemical studies also require the reduction of the optimization complications in order to be of use in real-time screening of melamine. Notably, in the main stream these reports displayed complications in their optimization along with the great recoveries and linearity in melamine detection, as summarized in Table 3.

**Table 3.** Summary of methods, instruments, optimization status, time, linear ranges, detection limits (LODs), recoveries and relative standard deviations (RSDs) of Nanorods and Nanotubes-based probes in melamine detection.

| Nanotube/Nanorod/ Nanowire Arrays | Method of Detection | Instruments Employed | Optimization Status and Time Required | Linear Range | LOD | Recoveries of Spiked Samples | RSDs | Ref. |
|---|---|---|---|---|---|---|---|---|
| Bis(8-quinolinolato) zinc (II) complex nanorod arrays | Fluorescence | PL spectrometer | Moderate and NA | 39.6 nM–238 n M | NA | NA | NA | [196] |
| ZnO/Au composite nano arrays | SERS | Raman spectrometer | Complicated and >0.5 min | 100 μM–10 nM | 10 nM | NA | NA | [197] |
| Ag-nanoparticle-modified single Ag nanowire | SERS | Raman spectrometer | Complicated and 60 min | 10 nM–22 μM | 10 nM | NA | NA | [198] |
| hydroxyapatite/carbon nanotubes | Electrochemical | Cyclic voltammeter | Complicated and 9 min | 10–350 nM | 1.5 nM | 98.5–102.5% | 1.32–2.58% | [200] |
| glassy carbon electrode coated with a multi-wall carbon nanotube/chitosan | Electrochemical | Cyclic voltammeter | Complicated and 20 min | 9.9–190 nM | 3 nM | 104.8% | NA | [201] |
| Vertically aligned monolayer of Aunanorods | SERS | Raman spectrometer | Complicated and 90 min | NA | ~0.9 fM | NA | NA | [202] |
| Single gold nanoparticles decorated silver/carbon nanowires | SERS | Raman spectrometer | Complicated and 60 min | 0.1–220 μM | 0.1 μM | NA | NA | [203] |
| Au nanorods coupled with Ag nanoparticles | SERS | Focus ion beam and Raman spectrometer | Complicated and >0.5 min | 1 mM–1 pM | 1 pM | NA | NA | [204] |
| disordered silver nanowires membrane | SERS | Raman spectrometer | Moderate and >0.5 min | 7.93 μM–0.79 mM | NA | NA | NA | [205] |
| Ag nanoparticles surrounding triangular nanoarrays | SERS | Raman spectrometer | Complicated and NA | 0.5–500 μM | 10 μM | NA | NA | [206] |
| ZnGa$_2$O$_4$ Nanorod Arrays Decorated with Ag Nanoparticles | SERS | Raman spectrometer | Complicated and >1 min | 0.1–100 μM | 0.1 μM | NA | NA | [207] |

**Table 3.** *Cont.*

| Nanotube/Nanorod/ Nanowire Arrays | Method of Detection | Instruments Employed | Optimization Status and Time Required | Linear Range | LOD | Recoveries of Spiked Samples | RSDs | Ref. |
|---|---|---|---|---|---|---|---|---|
| Au nanorod arrays fabrication using a focused gallium (Ga) ion beam | SERS | Focus ion beam and Raman spectrometer | Complicate & >1 min | 100 µM–1 pM | 1 pM | NA | NA | [208] |
| [Ru (bpy)$_3$]$^{2+}$-doped Si NPs/multi-walled carbon nanotubes/Nafion composite electrode | Electro-chemiluminescence | Multifunction chemiluminescence detector | Complicate &NA | 0.1 µM–0.5 pM | 0.1 pM | 99.7–102% | 1.1–3.1% | [209] |
| Ag nanorod (Ag NR) array | SERS | Raman spectrometer | Moderate & >0.5 min | 15.86 µM–1.59 mM | 7.1 µM | 89.7–93.3% | 1.08–2.02% | [210] |
| Flexible silicon nanowires | SERS | Raman spectrometer | Moderate & >0.5 min | 79 pM – 0.79 mM | 2.5 nM | NA | NA | [211] |
| CarbonNitride Nanotubes | Molecular Imprinted Voltammetry | Cyclic voltammeter | Moderate & 30 min | 0.1–5 nM | 10 pM | 98.68–102.94% | NA | [212] |
| Molybdenum Oxide-nanowires @ Au | SERS | Raman spectrometer | Complicate & >24 h | 0.79 nM–0.79 mM | 0.792 nM | NA | NA | [213] |
| Zinc oxide Nanowires decorated with Ag NPs | SERS | Raman spectrometer | Moderate & 60 min | 12 µM–76 µM | NA | NA | NA | [214] |
| Ag nanorod from polymeric silver cyanide | SERS | Raman spectrometer | Moderate & NA | 1 mM–1 pM | NA | NA | NA | [215] |
| Ag nanoparticles decorated Cu(OH)$_2$ nanoneedle | SERS | Raman spectrometer | Complicate & NA | NA | 0.792 nM | NA | NA | [216] |
| Ag NPs decorated Zinc Oxide/ Siliconhetrostructured nanomace Arrays | SERS | Raman spectrometer | Complicate & >0.5 min | 10 µM–0.1 nM | 10 fM | NA | NA | [217] |

NA = Not available; mM = millimole; µM = micromole; nM = nanomole; pM = picomole; fM = femtomole; min = minutes; Hrs = Hours.

The majority of nanowires/nanorods/nanotubes-based composite arrays engaged in melamine detection were attributed to the SERS effect of the substrate fabricated and subjected to the analysis [197–199,202–208,210,211,213–217]. In contrast, some tactics, such as fluorescence [196], electrochemical [200,201], electrochemiluminescence (ECL) [209], and voltammetry [212] were also employed in the discriminative assay of melamine. From the table, one can conclude that nanorod/nanowires/nanotubes arrays can be engaged as SERS substrates in making successful assays for the melamine detection in milk samples. However, much focus is still required to reduce the optimization complications and to enhance the linear range of detection and recoveries.

## 8. Other Nanostructures in Melamine Discrimination

Few research units have demonstrated the melamine determination through diverse nanostructures, as illustrated next. Rajkumar and co-workers presented the diverse nanostructured Ag NPs deposited on silicon substrates via one-step galvanic displacement method, which was further engaged in the SERS-based detection of melamine [218]. As shown in Figure 21, the SERS peak at 685 cm$^{-1}$ distinguishes the presence of melamine between 1 mM to 0.1 nM. This work showed a LOD of melamine of ~10 nM, and hence established its affordability in real-time monitoring of melamine. Using this SERS-based sensing approach, Zhang et al. presented the sandwich-type nanostructured substrate consisted of a probe molecule sandwiched between silver nanoparticles (SNPs) and silver nanoarrays for the selective assay of melamine [219]. The above sandwich nanostructure was fabricated on porous anodic aluminum oxide (AAO) by means of electrodepositing technique. The probe showed linearity to melamine between 1 mM to 1 nM and displayed a sensitivity up to 1 nM. However, the authors did not provide any clear information of the probe molecules. Hence, extensive melamine detection using the probe in dairy products is still questionable.

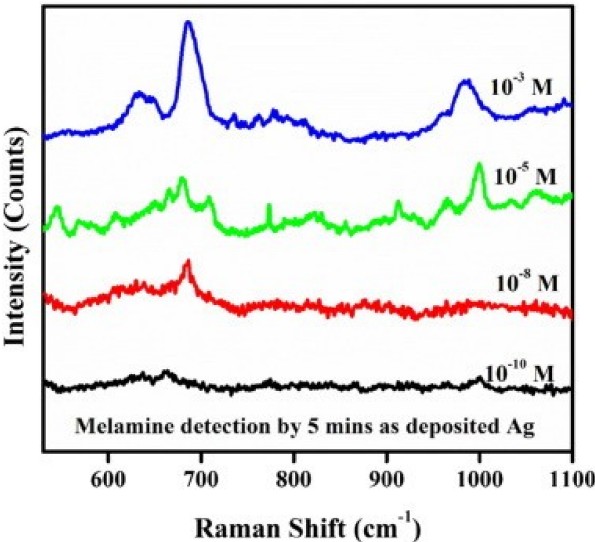

**Figure 21.** SERS Spectra of Melamine with different concentrations for 5 min Ag deposited samples (reproduced with permission from Reference [218]).

For the determination of melamine using electrochemical sensing, Cadmium doped antimony oxide nanostructures (CAO-NSs) were employed to modify the glassy carbon electrode [220]. The CAO-NSs/GCE demonstrated the linearity in melamine assay from 0.05 nM–0.5 mM with a LOD of 14 ± 0.05 pM. Recoveries by this method in milk samples were established between 97.3~103.5% with 1.9~3.5% RSDs. This electrochemical method appears to be one of the best techniques, and hence can be employed for the discrimination of melamine in dairy stuffs. Ibupoto's research utilized the succinic acid-functionalized copper oxide nanostructures to modify the GCE and then applied in the electrochemical discovery of melamine [221]. The GCE/CuO-NSs/nafion detected the melamine

linearly between 100 pM to 5.6 nM with a LOD of 10 pM. Recoveries by this nanostructure were 98~99% with 0.31~0.83% RSDs. Due to the excellent selectivity to melamine, the modified electrode can be validated as a great innovation in the melamine sensing.

Guo et al. presented a SERS-based melamine quantification in milk samples by employing the hollow gold nanospheres (HGNs) on glass wafers via electrostatic interaction as illustrated in Figure 22 [222]. Here, the HGNs displayed a strong SERS enhancement to melamine due to its ability to confine the electromagnetic fields around the pinholes in hollow shells. The hollow gold chip discovered the melamine linearly between 0 to 793 μM with a LOD of 7.93 μM. This method was validated in milk samples, but it can only be considered as an addition to those SERS-mediated detection tactics. Similar to the SERS methods, Jean and collaborators proposed the optical sensing of melamine by means of Ag decorated silica nanoparticles ($SiO_2$@Ag nanospheres) [223]. The optical sensor can sense the melamine at nM level with great sensitivity between 793 nM to 7.93 mM. Silver nanoparticles coated amino modified polystyrene microspheres (PS-$NH_2$/Ag NPs) were demonstrated in the assay of melamine by Zhao and collaborators [224]. The melamine detection displayed linear regression from 1 mM to 10 nM at 698 cm$^{-1}$ and have a LOD of 19 nM. Moreover, this work was demonstrated in milk powder samples without any sample pretreatment steps. Hence, it can be engaged in real-time inspection of melamine contamination.

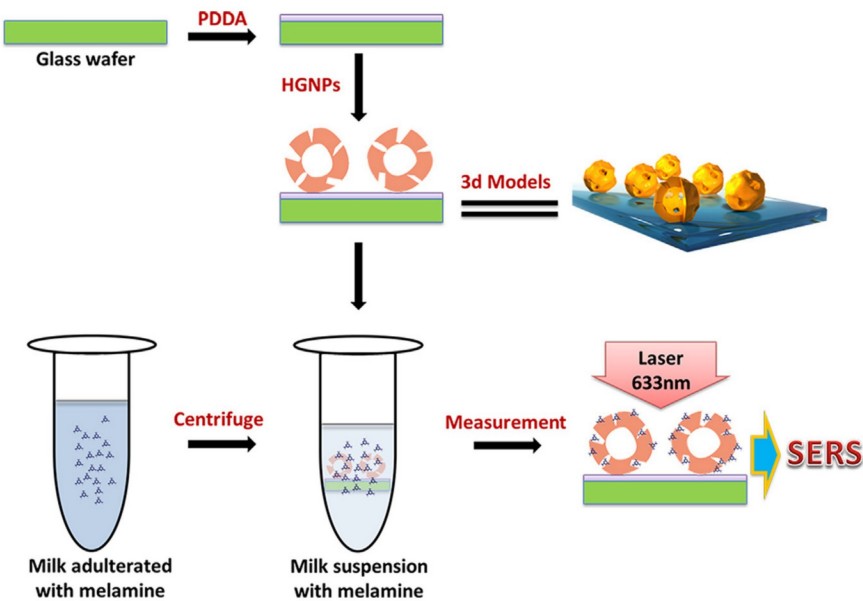

**Figure 22.** Procedures of hollow gold (HG) chip fabrication and detection procedures of melamine from real milk sample by HG chip (reproduced with permission from Reference [222]).

The SERS-based determination of melamine was further established using Ag@$SiO_2$ nanocubes by Su and Hwang's research groups [225]. The above nanocube materials displayed the linearity to melamine from 0.5 to 7.93 μM at 684 cm$^{-1}$ with a LOD of 0.48 μM. Moreover, Ag@$SiO_2$ nanocubes also showed the linear melamine detection in milk samples, which was between 2.46 to 39.6 μM with a LOD of 1.35 μM. Recoveries in spiked milk samples were found as 94.86 to 99.87% with 11.11~17.11% RSDs. This method can be categorized as a suitable SERS method. In a similar fashion, Qin et al. reported the hollow nanocubes made of Ag–Au alloys for SERS-based recognition of melamine [226], in which the sensitivity to melamine was established at 701 cm$^{-1}$ at a concentration as low as 10 nM. Even though this work is elaborated, still much focus is required to verify its potentiality in milk products. Chen et al. demonstrated the applicability of urchin-like $LaVO_4$/Au composite microspheres towards the melamine discovery via SERS responses at 682 cm$^{-1}$ with linearity between 10 μM to 1 nM [227]. LOD of melamine recognition by this tactic was estimated to be 1 nM, and hence can be categorized as one of the SERS methods. However, the applicability of this probe is still in need of verification.

A portable multi-channel sensing device comprised of Au nano-urchins was described by Huang and Chen collaborators via localized surface plasmon resonance (LSPR) at 680 nm [228]. In the presence of melamine the LSPR peak at 680 nm was enhanced and exhibited linear regression between 0.1 to 1 μM with a LOD of 18 nM. Potentiality of the above research needs to be validated in real samples for future applications. As shown in Figure 23, Nguyen et al. presented the use of sharp-edged gold nanostar (Au NSs) substrates via SERS response at 714 cm$^{-1}$ for on-site determination of melamine in infant formula and chocolate [229]. They compared the potentiality of Au NSs with the spherical Au NPs. In the infant formula, the Au NSs and Au NPs showed the linear regressions of 0.79–793 nM and 79.3 nM–39.6 μM with LODs of 79.3 nM and 0.79 μM, respectively. Similarly, Au NSs and Au NPs showed the linear regressions of 7.93 nM–19.8 μM and 7.93–79.3 μM in chocolate mixture with LODs of 0.79 μM and 79.3 μM, correspondingly. Due to the authenticated real-time application, this nanostructured probe can be used for efficient monitoring of melamine in dairy products.

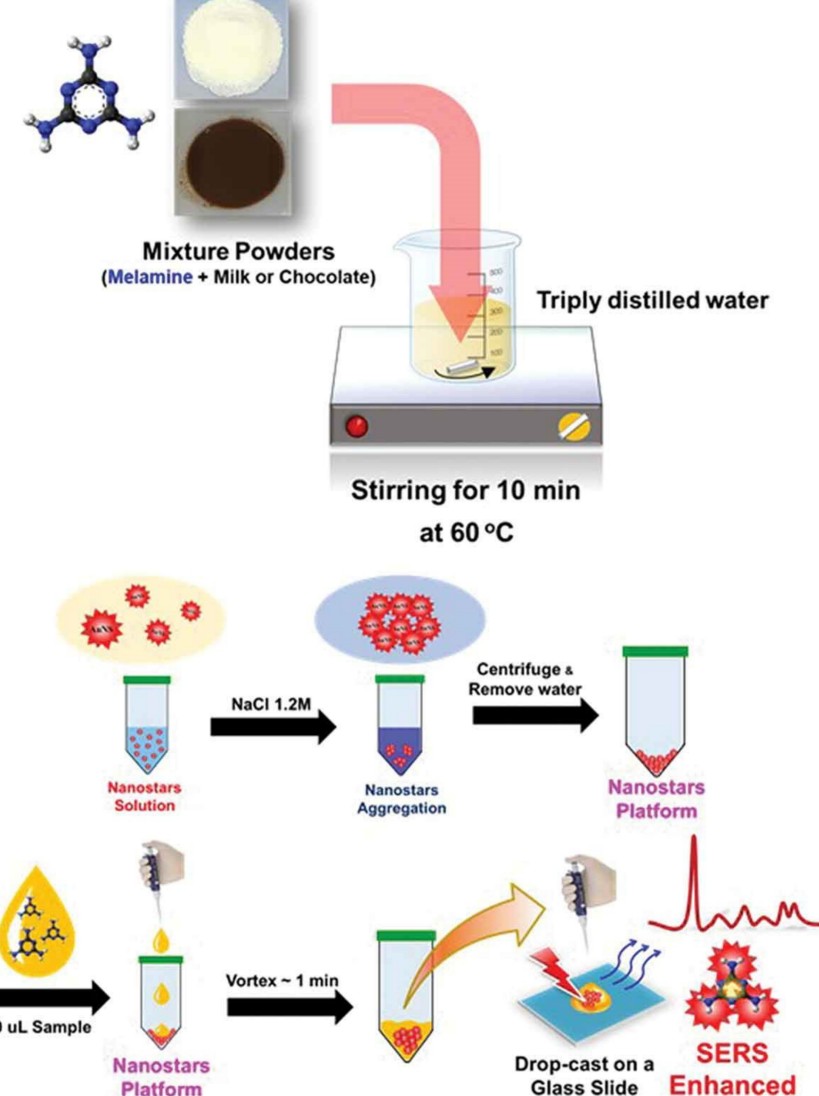

**Figure 23.** Experimental schemes for detecting melamine in powdered infant formula and chocolate. On-site detection of sub-mg/kg melamine could be achieved using Raman spectroscopy within a few minutes (reproduced with permission from Reference [229]).

As a different nanostructured material, gold coated zinc oxide nanonecklaces (ZnO NN) arrays were utilized towards melamine sensing by He and co-workers [230]. These ZnO NN arrays detected the melamine via SERS signal at 683 cm$^{-1}$ with a LOD of 10 μM. Between 10 mM to 1 μM, the probe

displayed the linearity, but the sensitivity towards melamine still needed to be improved for real-time application. In this track, gold-sputtered Blu-ray discs (BD-R) were employed to discover melamine by Nieuwoudt et al. via SERS signal at 683 cm$^{-1}$ [231]. From concentrations of melamine of 0.79 nM to 7.93 mM the SERS signals were observed with a LOD of 555 nM. However, although the above method shows the lowest detection range, the reliability is not well demonstrated. Using the SERS-based tactics, Cao and co-workers demonstrated the trace level melamine sensing by self-assembly of silver nanostructures on carbon-coated copper grids [232]. By means of surfactant-free and ultrafast self-assembly over carbon-coated copper TEM grids, well defined silver structures comprised of nano-flowers (NFs), planar nanospheres (NSs), and nano-dendrites (NDs) were fabricated and applied in the SERS-based melamine recognition studies. At 688 cm$^{-1}$, the SERS signal of melamine displayed linearity between 10 pM to 1 µM with a LOD of 10 pM. The grids used in this method can be re-used after exposure to air for six months and can still achieve the same performance. Therefore, this method is authenticated as one of best SERS-based techniques for real-time inspection of melamine in food stuffs.

Similar to the self-assembly of nanostructures, melamine and its derivatives can produce nanostructures via H-bonding with suitable candidates or by self-assembly [233–239]. Functional materials can form during these process, which may find their use in melamine detection in near future. For example, the H-bonded hydrogel formed by melamine with a molecule Nap-FFYGK-CA was reported as a good tactic for the assay by Yang and Chen collaborators [234]. The above work was demonstrated in milk and urine samples, and can thus be employed as a valid scheme for melamine discrimination in future.

## 9. Advantages and Limitations

The nanomaterial-based melamine assays have several advantages as well as limitations as stated below.

(1) Many novel designs from nanomaterials have been proved their effectiveness towards melamine quantifications in milk and food stuffs via dissimilar detection methodologies, which allow the modern world to remain healthy and safe.

(2) The majority of nanomaterial-based melamine sensors have the lowest detection limits (femtomolar to nanomolar) with excellent linearities, and hence can determine the melamine at low concentrations in dairy products.

(3) Diverse tactics in melamine detection have been employed by nanomaterials, which allow them to be utilized towards the inspection of dissimilar contaminated samples. For example, fluorescence and colorimetric tactics of NPs and NCs can help to identify the melamine in solution by the naked eye. On the other hand, SERS and electrochemical techniques may recognize melamine through specific signals from solution and powders.

(4) The metal ion-mediated detection probes for melamine can act as dual-mode sensors to metal ions and melamine in environmental water samples and dairy products, respectively.

(5) The cost-effectiveness of the majority of nanomaterial-based assay tactics in melamine detection appears to be low, and hence can be used in real-time monitoring of melamine.

(6) Design of nanoparticles for melamine determination is limited by the electrostatic forces and functional units presented over their surface, which must show the tendency to form H-bond with melamine or to coordinate/complex with melamine. However, to identify such functional units, extended research work is required.

(7) For metal nanocluster-based assay of melamine, the quantum yield should be improved by modifications with suitable groups [240], otherwise the LODs on melamine quantification will not be improved.

(8) Nanocomposite-based discrimination of melamine is restricted by the compositions of the mixtures which require essential property. Moreover, the development of nanocomposite with essential properties are still time consuming. Therefore much effort is needed in their optimization.

(9) The nanorods/nanowires/nanotubes-based melamine sensing approaches are limited by their melamine-capture ability. It means that not all of the NWs/NRs/NTs can detect the melamine. Moreover, the main stream of these arrays-based assays is complicated in terms of optimizations. Hence much focus is needed to overcome the difficulties and to enhance the reliability.

(10) To identify the mechanisms behind the melamine sensing processes with nanomaterials, sophisticated instruments, such as TEM, Raman spectroscopy, and cyclic voltammetry are essential, which will affect the cost and time in developing melamine sensing techniques. Therefore, much anticipation is expected for these tactics.

## 10. Conclusions and Perspectives

In this review, we have summarized the nanomaterial-based sensors for illegal food contaminant melamine. Note that this review covers the melamine assays using nanomaterials, such as carbon dots, quantum dots, nanocomposites, nanocrystals, nanoclusters, nanoparticles, nanorods, nanowires, and nanotubes. Moreover, diverse mechanisms, including fluorescence resonance energy transfer (FRET), aggregation, inner filter effect, surface-enhanced Raman scattering (SERS), and self-assembly in melamine determination are discussed in detail. Wherein, Au NPs- and Ag NPs-based colorimetric sensing via aggregation or H-bonding appears to be a promising strategy among the majority of the reports. On the other hand, many scientists have developed diverse nanomaterials, which address the melamine contamination through SERS tactics. Apart from the above methods, a few reports have also covered FRET, CRET, Sonoluminescence, and Chemiluminescence, etc.

However, the following nanomaterial-based melamine assays and mechanisms are still missing, which need to be established in the future.

(1) The C-dots-based assay of melamine is not totally innovative, and hence much attention is required.

(2) Reports on Pt and bi-metallic nanoclusters towards melamine sensing are insufficient. More researchers must devote themselves to this research area.

(3) The mechanisms of a few nanocomposite-based melamine recognitions are not entirely clear, and hence should be further investigated in the future.

(4) Studies on band-gap properties of the nanorods/nanowires/nanotubes-based sensors may help to interpret the mechanisms involved, which should be evaluated in the near future.

(5) Researchers should be encouraged to investigate other nanostructures, such as nanocubes, nanostars, nanoflowers, and nanocrystals towards melamine assays.

Even though developing nanomaterial-based melamine assays requires sophisticated optimization procedures and delicate instruments, they provide a breakthrough in food contamination assay and allow the world to sustain a safe and healthy environment. Moreover, researchers are devoted to the development of nanomaterials, which may revolutionize the current health and food industries

**Author Contributions:** This review written by M.S and K.W.S with equal contributions.

**Conflicts of Interest:** The authors declare no conflicts of interest.

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
