# Peer review of "Review on Nanomaterial-Based Melamine Detection"

_chemosensors, doi:10.3390/chemosensors7010009_

Round 1
Reviewer 1 Report
Extensive English editing by professional English editor is required for publication. Typos and grammatical mistakes make it hard for readers to follow. (examples from the page 1: line 11: "China government" to "Chinese government", line 16: "quiet" to "quite")
As a review paper, I recommend inserting outline for easier navigation of the contents.
Locate Figure 3 after the paragraph, not in between the sentence.
Please unify concentration units of the linear range and LOD in the Tables for easier comparison between the techniques.
Font size is different in page 46.
Compare approximate time and complexity for each measurements in the Table as one of the main advantages over conventional methods is stated as simplicity in the introduction.
The authors compared the detection technologies using nanomaterials used (e.g. Table 1 is AuNP and Table 2 is AgNP). However, wouldn't it be better to compare between different detection method such as Table 1 for Colorimetric/Fluorescence and Table 2 for SERS? Then, you can compare the pros and cons of different detection methods in the conclusion to suggest optimal choices by different situations for the readers.
Author Response
Answers to the Comments of Reviewer 1
Extensive English editing by professional English editor is required for publication. Typos and grammatical mistakes make it hard for readers to follow. (examples from the page 1: line 11: "China government" to "Chinese government", line 16: "quiet" to "quite")
“Author Reply”
We are very thankful to the reviewer, for providing the valuable comments to improve the standard of our review article. As per reviewer’s comments, we have corrected those lingual deficiencies with the consultation of native English speaker.
As a review paper, I recommend inserting outline for easier navigation of the contents.
“Author Reply”
As per reviewer’s suggestion outline has been included in the revised manuscript.
Locate Figure 3 after the paragraph, not in between the sentence.
“Author Reply”
Figure 3 has been located properly in the revised format.
Please unify concentration units of the linear range and LOD in the Tables for easier comparison between the techniques.
“Author Reply”
As suggested, concentration units throughout the manuscript has been unified to Molar level.
Font size is different in page 46.
“Author Reply”
All font sizes were unified in current format.
Compare approximate time and complexity for each measurements in the Table as one of the main advantages over conventional methods is stated as simplicity in the introduction.
“Author Reply”
Instruments employed, optimization status and time required for melamine assays were included in the Tables, which suggest the effectiveness and cost requirement.
The authors compared the detection technologies using nanomaterials used (e.g. Table 1 is AuNP and Table 2 is AgNP). However, wouldn't it be better to compare between different detection method such as Table 1 for Colorimetric/Fluorescence and Table 2 for SERS? Then, you can compare the pros and cons of different detection methods in the conclusion to suggest optimal choices by different situations for the readers.
“Author Reply”
At this stage, it is not far enough present the Tables as recommended by the reviewer. Because, only few references have SERS effect in sensing than colorimetric studies. Moreover, some probes have both responses. However, currently we include Instruments employed, optimization status and time required in those tables, which will give idea for readers regarding innovative tactics.
Reviewer 2 Report
The manuscript describes the use of nanomaterial based melamine detection. Although the manuscript is comprehensive in its review of this subject and would be of interest to the research community, especially the Asian research community where there is a lot of research efforts in this area. However, there are major flaws with this manuscript. The authors have not been careful in the proof reading of this manuscript and there are numerous grammatical errors littered throughout the manuscript. I recommend that the manuscript be carefully proof read before further consideration. The layout of the review is not adequate which can make it difficult to read. For instance the authors have set out the sections in terms of type of nanomaterial which is problematic as the examples given in each section tend to overlap with other areas.
I think it would be better and clearer for the reader if the authors discussed the methods for the detection of melamine in terms of detection method. i.e Fluorescence, absorbance, electrochemical etc.
I therefore cannot recommend publication without a major revision of the manuscript. In addition, a number of points have been raised below which should be addressed.
Line 9: a threat
Line 10: in babies
Line 11: Chinese Government
Line 16: quite promising
Figure 1: For 2017 the number of publications decreased. The authors should comment on why this is. From the graph, there is a suggestion that there is now less interest in developing melamine sensors.
The authors report the LOD and concentration ranges in different units i.e uM, ng ml, ppm. All LODs and calibration ranges within the manuscript should be converted to the same units to make comparison between methods easier. I would recommend changing to Molarity.
Line 198: This should be mentioned in the Introduction.
Line 218: The authors should define Hapten and give more detail on the design of this sensor.
Line 238: The authors should describe the drawbacks of antibody based sensors for melamine detection.
Figure 7 appears to be upside down.
No citation to figure 8
Figure 13 is too small and needs to be increased in size.
Figure 24 has a water mark of manuscript written on it.
Other than Milk and dairy products, have these sensors been used to test melamine in other food matrices. The authors should discuss this in more detail.
The list of advantages and disadvantages is vague and isn’t really clear. The authors should be clearer.
Line 1192: What does this mean? “Nanocomposite based discrimination of melamine is restricted by the compositions of the mixtures, which requires further optimization”. Surely the whole point is that a nanocomposites can be tailored to contain the properties that you want.
Author Response
Answers to the Comments of Reviewer 2
The manuscript describes the use of nanomaterial based melamine detection. Although the manuscript is comprehensive in its review of this subject and would be of interest to the research community, especially the Asian research community where there is a lot of research efforts in this area. However, there are major flaws with this manuscript. The authors have not been careful in the proof reading of this manuscript and there are numerous grammatical errors littered throughout the manuscript. I recommend that the manuscript be carefully proof read before further consideration. The layout of the review is not adequate which can make it difficult to read. For instance the authors have set out the sections in terms of type of nanomaterial which is problematic as the examples given in each section tend to overlap with other areas.
“Author Reply”
We sincerely thank the reviewer for the valuable suggestions and comments. We have corrected the lingual deficiencies with the consultation of native English speaker. An outline has been included in the revised manuscript with proper layout. We agree the overlapping of few references, however, those references were included in to specific sections based on the statement of authors or from the highlighted results.
I think it would be better and clearer for the reader if the authors discussed the methods for the detection of melamine in terms of detection method. i.e Fluorescence, absorbance, electrochemical etc.
“Author Reply”
Such discussions based on methods led to overlapping of diverse nanomaterials reports, that is why we described based on nanomaterials. Further, method based thoughts also violates the scope of the manuscript.
Line 9: a threat
Line 10: in babies
Line 11: Chinese Government
Line 16: quite promising
“Author Reply”
Those typo errors were rectified in the revised manuscript.
Figure 1: For 2017 the number of publications decreased. The authors should comment on why this is. From the graph, there is a suggestion that there is now less interest in developing melamine sensors.
“Author Reply”
The attention to develop melamine detection probes are still in need as seen the No of Publications in 2018.
The authors report the LOD and concentration ranges in different units i.e uM, ng ml, ppm. All LODs and calibration ranges within the manuscript should be converted to the same units to make comparison between methods easier. I would recommend changing to Molarity.
“Author Reply”
As per suggestion, concentration units throughout the manuscript has been unified to Molar level.
Line 198: This should be mentioned in the Introduction.
Line 218: The authors should define Hapten and give more detail on the design of this sensor.
Line 238: The authors should describe the drawbacks of antibody based sensors for melamine detection.
Figure 7 appears to be upside down.
No citation to figure 8
Figure 13 is too small and needs to be increased in size.
Figure 24 has a water mark of manuscript written on it.
“Author Reply”
Those issues were fixed in the current manuscript. Further, few figures has been omitted to shorten the document.
Other than Milk and dairy products, have these sensors been used to test melamine in other food matrices. The authors should discuss this in more detail.
“Author Reply”
Since it is essential to reduce the length of the manuscript as suggested by other referees. Currently, we did not supplement such discussions.
The list of advantages and disadvantages is vague and isn’t really clear. The authors should be clearer. Line 1192: What does this mean? “Nanocomposite based discrimination of melamine is restricted by the compositions of the mixtures, which requires further optimization”. Surely the whole point is that a nanocomposites can be tailored to contain the properties that you want.
“Author Reply”
Clarification on advantages and disadvantages were delivered in the revised manuscript.
Reviewer 3 Report
The authors reviewed the recent progress on the nanomaterial-based melamine sensors. Even though the literatures reviewed in this manuscript are quite comprehensive, the layout of review in terms of type of nanomaterial is not appropriate since many sensors have the quite similar sensing mechanisms regardless of the type of nanomaterials. It is suggested to organize the review according to different mechanisms (Hg2+ mediated methods, aggregation, fluorescence quenching, fluorescence enhancement….) or signals (colorimetric, fluorescent, and electrochemical). In addition, the writing is poor and needs largely improved. There are numerous typos and grammatical errors throughout the manuscript. I recommend rejection and resubmission. The additional comments are list below.
(1) Please unify concentration units of the linear range and LOD throughout the whole manuscript including tables for easier comparison between different techniques.
(2) The cost, assay time, sensitivity, and specificity are the four most important factors to consider for choosing analytical methods. Please include the cost (compared to immunoassay) and assay time, and applicability for real sample in the Tables.
(3) It is suggested to summary methods in the tables according to different mechanism or signals instead of the type of nanomaterials. In addition, the specific mechanism should be provided to differentiate different methods with the same type of signal output. For example, in Table 1, there are many citrate stabilized Au NPs-based colorimetric sensors. What are their differences? Can these literatures be combined into one since the table is too large?
(4) Some figures are not necessary. One or two sentences are enough to describe these figures, such as Figure 1 and 2.
(5) Figure 7 is upside down.
(6) The whole paper is lack of in-deep discussion and concise summary, which makes this paper unnecessary long and less readable. The similar works should be summarized together and only one of them should be described in detail. The advantages and disadvantages among these similar methods should be discussed.
Author Response
Answers to the Comments of Reviewer 3
We are grateful to the reviewer, for providing the valuable comments to improve the standard of our manuscript.
The authors reviewed the recent progress on the nanomaterial-based melamine sensors. Even though the literatures reviewed in this manuscript are quite comprehensive, the layout of review in terms of type of nanomaterial is not appropriate since many sensors have the quite similar sensing mechanisms regardless of the type of nanomaterials. It is suggested to organize the review according to different mechanisms (Hg2+mediated methods, aggregation, fluorescence quenching, fluorescence enhancement….) or signals (colorimetric, fluorescent, and electrochemical). In addition, the writing is poor and needs largely improved. There are numerous typos and grammatical errors throughout the manuscript.
“Author Reply”
Discussions based on mechanisms led to overlapping of diverse nanomaterials reports, that is why we described based on type of nanomaterials. Further, method based thoughts also violates the scope of the manuscript. Wehave corrected the lingual deficiencies with the consultation of native English speaker.
(1) Please unify concentration units of the linear range and LOD throughout the whole manuscript including tables for easier comparison between different techniques.
“Author Reply”
As recommended by the reviewer, concentration units throughout the manuscript has been unified to Molar level.
(2) The cost, assay time, sensitivity, and specificity are the four most important factors to consider for choosing analytical methods. Please include the cost (compared to immunoassay) and assay time, and applicability for real sample in the Tables.
“Author Reply”
Instruments employed, optimization status and time required for melamine assays were included in the Tables, which suggest the effectiveness and cost requirement. Without sensitivity, and specificity, research cannot be publicized, hence they were omitted in the Tables.
(3) It is suggested to summary methods in the tables according to different mechanism or signals instead of the type of nanomaterials. In addition, the specific mechanism should be provided to differentiate different methods with the same type of signal output. For example, in Table 1, there are many citrate stabilized Au NPs-based colorimetric sensors. What are their differences? Can these literatures be combined into one since the table is too large?
“Author Reply”
The included optimization status and time required for melamine assays in Tables witnessed the difference between those similar citrate stabilized Au NPs. The main idea of this paper to emphasize the literatures on nanomaterials directed melamine assays. Wherein, similarly functionalized nanoparticles may have diverse particle sizes and detect melamine at varied time intervals, which will give awareness to upcoming researchers.
(4) Some figures are not necessary. One or two sentences are enough to describe these figures, such as Figure 1 and 2.
“Author Reply”
Few figures has been omitted and many sentences were shortened in revised the document.
(5) Figure 7 is upside down.
“Author Reply”
Figure 7, fixed properly in the current manuscript.
(6) The whole paper is lack of in-deep discussion and concise summary, which makes this paper unnecessary long and less readable. The similar works should be summarized together and only one of them should be described in detail. The advantages and disadvantages among these similar methods should be discussed.
“Author Reply”
Discussions are enhanced with concise summary in the revised format. The advantages and disadvantages among these similar methods has been clarified in the revised paper.
Round 2
Reviewer 1 Report
English should be further improved. I suggest second revision from professional English editor.
Author Response
Manuscript has been smoothed by English professional.
Reviewer 2 Report
Most of the comments have been addressed. However the authors should address the following point and further proof read the manuscript.
What is the difference between a nanocrystal and a nanoparticle? In my opinion, they are both the same. Can the authors give a definition for both?
Author Response
Polycrystalline materials with a crystallite size of few nanometers are defined as nanocrystals. In other words, "nanocystal" refers to the structure of a small piece of condensed matter (not necessarily individualised), which fill the gap between amorphous materials without any long range order and conventional coarse-grained materials.
In contrast, nanoparticle is a particulate material of a nano size. It can be made up of nanocrystal or amorphous materials in nature. In precise, "nanoparticle" only refers to the size of a small, individualised, matter, whether crystalline or not.
Reviewer 3 Report
The authors have revised the manuscript according to the reviewer' suggestions.
Author Response

(The authors gave the same response as above.)
